# GARNET: A Spectral Approach to Robust and Scalable Graph Neural Networks

## Abstract

Graph neural networks (GNNs) have been increasingly deployed in various applications that involve learning on non-Euclidean data. However, recent studies show that GNNs are vulnerable to graph adversarial attacks. Although there are several defense methods to improve GNN adversarial robustness, they fail to perform well on low homophily graphs. In addition, few of those defense models can scale to large graphs due to their high computational complexity and memory usage. In this paper, we propose GARNET, a scalable spectral method to boost the adversarial robustness of GNN models for both homophilic and heterophilic graphs. GARNET first computes a reduced-rank yet sparse approximation of the adversarial graph by exploiting an efficient spectral graph embedding and sparsification scheme. Next, GARNET trains an adaptive graph filter on the reduced-rank graph for node representation refinement, which is subsequently leveraged to guide label propagation for further enhancing the quality of node embeddings. GARNET has been evaluated on both homophilic and heterophilic datasets, including a large graph with millions of nodes. Our extensive experiment results show that GARNET increases adversarial accuracy over state-of-the-art GNN (defense) models by up to $9.96\%$ and $18.06\%$ on homophilic and heterophilic graphs, respectively.

## 1 Introduction

Recent years have witnessed a surge of interest in graph neural networks (GNNs), which incorporate both graph structure and node/edge attributes to produce low-dimensional embedding vectors that maximally preserve graph structural information (Hamilton, 2020). GNNs have achieved promising results in various real-world applications, such as recommendation systems (Ying et al., 2018), self-driving car (Casas et al., 2020), protein structure predictions (Senior et al., 2020), and chip placements (Mirhoseini et al., 2021). However, recent studies have shown that adversarial attacks on graph structure accomplished by inserting, deleting, or rewiring edges in an unnoticeable way, can easily fool GNN models and drastically degrade their accuracy in downstream tasks (e.g., node classification) (Zügner et al., 2018; Zügner & Günnemann, 2019).

In literature, there are several attempts to defend GNNs against graph adversarial attacks. Entezari et al. (2020) first observed that graph adversarial attacks mainly affect high-rank properties of the graph; consequently, a low-rank graph should be first constructed by performing truncated singular value decomposition (SVD) on the graph adjacency matrix, which can then be exploited for training a robust GNN model. Later, Jin et al. (2020) proposed Pro-GNN to jointly learn a new graph and its robust GNN model with the low-rank constraints imposed by the graph structure. However, such low-rank approximation methods involve dense adjacency matrices during the GNN training stage, which will lead to quadratic time and space complexity, prohibiting their applications in large-scale graph learning tasks. Another line of research aims at enhancing GNN robustness based on the graph homophily assumption, i.e., adjacent nodes in a natural graph tend to have similar attributes, while the graph attacks essentially insert adversarial edges by connecting nodes with dissimilar attributes. As a result, researchers proposed to compute attribute similarity scores between adjacent nodes as edge weights, and drop edges with small weights to increase graph homophily (Wu et al., 2019; Zhang & Zitnik, 2020). Although such approaches can successfully improve GNN robustness on high-homophily graphs, they may fail on low-homophily (i.e., heterophily) graphs (Zhu et al., 2020).

In this paper, we propose GARNET, a spectral method for constructing GNN models that are robust to graph adversarial attacks for both homophilic and heterophilic graphs. In addition, GARNET scales comfortably to large graphs due to its nearly-linear algorithm complexity. More concretely, GARNET consists of three major kernels: (1) reduced-rank approximation, (2) adaptive filter learning, and (3) adaptive label propagation. The reduced-rank approximation kernel first performs truncated SVD on graph adjacency matrix to obtain a few dominant singular values and their corresponding singular vectors that are further leveraged to construct a sparse yet reduced-rank graph adjacency matrix; the reduced-rank adjacency matrix can effectively mitigate the effects of adversarial attacks via connecting (disconnecting) nodes that are spectrally similar (dissimilar). The adaptive filter learning kernel aims to learn a polynomial graph filter whose coefficients are trainable; the learned graph filter can adapt to the homophilic/heterophilic properties of the underlying graph and thus will work effectively for both homophilic and heterophilic graphs. The adaptive label propagation stage will leverage the learned adaptive graph filter to guide the label propagation phase, which can further improve the adversarial accuracy by enhancing the quality of node representations.

We evaluate the proposed GARNET model on three high-homophily datasets: Cora, Citeseer, and Pubmed as well as two low-homophily datasets: Chameleon and Squirrel, under strong graph adversarial attacks such as Nettack (Zügner et al., 2018) and Metattack (Zügner & Günnemann, 2019) with various perturbation settings. Moreover, we further show the nearly-linear scalability of our approach on the ogbn-products dataset that consists of millions of nodes (Hu et al., 2020). Our experimental results indicate that GARNET largely improves adversarial accuracy over baselines in most cases. The major advantages of GARNET are summarized as follows:

• **GARNET is robust to graph adversarial attacks.** Exploiting the proposed reduced-rank approximation scheme allows GARNET to effectively filter out the noises potentially caused by adversarial attacks in the spectral domain. This immediately leads to substantial improvement (up to $18.06\%$) of adversarial accuracy when comparing with the state-of-the-art baselines under the strong graph attacks on various datasets.

• **GARNET is agnostic to graph homophily.** Unlike most existing defense models that do not work well on low-homophily adversarial graphs, GARNET leverages a trainable graph filter that can adapt to adversarial graphs with diverse levels of homophily. In addition, we theoretically demonstrate that the performance of the adaptive graph filter applied to an adversarial graph will be similar to the performance achieved on a clean graph.

• **GARNET is scalable to large graphs.** GARNET has a nearly-linear runtime/space complexity and thus can scale comfortably to very large graph data sets with millions of nodes. We have conducted experiments on the ogbn-products dataset that contains 2 million nodes and 60 million edges, which is over $100\times$ larger than the largest adversarial graph ever considered in the prior arts.

## 2 RELATED WORK

GNNs have received an increasing amount of attention in recent years due to its ability of learning on non-Euclidean (graph) data. In contrast to developing powerful GNN models on natural graph data, there is an active body of research focusing on adversarial attacks as well as defenses for GNNs. We summarize some of the recent efforts for graph adversarial attacks and defenses as follows.

**Graph adversarial attacks** aim at degrading the accuracy of GNN models by perturbing the graph structure in an unnoticeable way. For instance, most existing graph adversarial attacks insert/delete edges while maintaining node degree distribution (Sun et al., 2018). The most popular graph adversarial attacks fall into the following two categories: (1) targeted attack, (2) untargeted attack. The targeted attacks attempt to mislead a GNN model to produce a wrong prediction on a target sample (e.g., node), while the untargeted attacks strive to degrade the overall accuracy of a GNN model for the whole graph data set. Dai et al. (2018) first formulate the targeted attack as a combinatorial optimization problem and leverages reinforcement learning to insert/delete edges such that the target node is misclassified. Zügner et al. (2018) propose another targeted attack called Nettack, which produces an adversarial graph by maximizing the training loss of GNNs. Zügner & Günnemann (2019) further introduce Metattack, an untargeted attack that treats the graph as a hyperparameter and uses meta-gradients to perturb the graph structure. It is worth noting that graph adversarial attacks have two different settings: poison (perturb a graph prior to GNN training) and evasion (perturb a graph after GNN training). As shown by Zhu et al. (2021), the poison setting is typically

more challenging to defend, as it changes graph structure that fools GNN training. Thus, in this work, we evaluate our model against both targeted and untargeted attacks under the poison setting.

**Graph adversarial defenses** attempt to enhance GNN performance on the perturbed graphs generated by adversarial attacks. Entezari et al. (2020) first observe that Nettack, a strong targeted attack, only changes the high-rank information of the adjacency matrix after graph perturbation. Thus, they propose to construct a low-rank graph by performing truncated SVD to undermine the effects of adversarial attacks. Jin et al. (2020) propose Pro-GNN that adopts a similar idea yet jointly learns the low-rank graph and GNN model. However, such low-rank approximation based methods produce dense adjacency matrices that correspond to complete graphs, which would limit their applications for large graphs. Another line of research strives to purify the adversarial graph by assigning edge weights. Specifically, Wu et al. (2019) propose to modify the edge weights by computing the Jaccard similarity score per edge based on node attributes, which is followed by Zhang & Zitnik (2020) that propose GNNGuard to learn node attribute similarity score per edge through a trainable linear layer. Nonetheless, such approaches assume that nearby nodes should have similar attributes (i.e., graph homophily assumption), which is not valid for heterophilic graphs that have adjacent nodes with dissimilar attributes (Zhu et al., 2020). In contrast to the prior arts, GARNET achieves highly robust yet scalable performance on both homophilic and heterophilic graphs under adversarial attacks by leveraging novel reduced-rank approximation and adaptive graph filtering schemes.

## 3 OUR APPROACH

Figure 1 gives an overview of our proposed approach, GARNET, which consists of three major phases. The first phase (reduced-rank approximation) constructs a reduced-rank yet sparse graph model by exploiting scalable truncated SVD and nearest-neighbor graph algorithms (Baglama & Reichel, 2005; Malkov & Yashunin, 2018). The second phase (adaptive filter learning) is the only phase that involves training, which outputs a learned graph filter that can adapt to the homophilic/heterophilic properties of the underlying graph. The last phase (adaptive label propagation) leverages the learned adaptive graph filter to guide the subsequent label propagation process. In the rest of this section, we will describe each of these three phases in more detail.

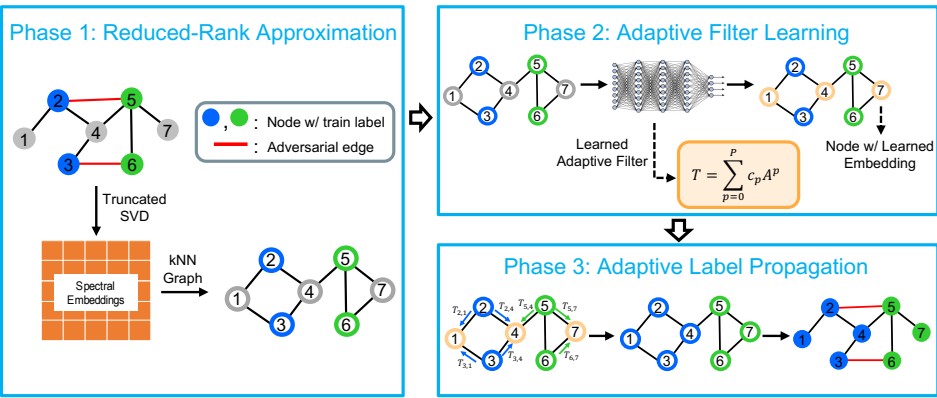

Figure 1: An overview of the three major phases of GARNET.

## 3.1 REDUCED-RANK APPROXIMATION (PHASE 1)

The adjacency matrices of many real-world graphs (e.g., social network and biological network) are naturally low rank and sparse, as the nodes typically tend to form communities and have a small number of neighbors (Zhou et al., 2013). As a result, graph adversarial attacks can be viewed as compromising these properties by inserting edges that connect nodes from different communities (Zügner et al., 2018; Zügner & Günnemann, 2019). Recently, Entezari et al. (2020) and Jin et al. (2020) have shown that the well-known graph adversarial attacks (e.g., Nettack and Metattack) are essentially high-rank attacks, which mainly change the high-rank spectrum of the graph when perturbing the graph structure, while the low-rank spectrum remains almost the same. We empirically confirm that the graph rank indeed increases under adversarial attacks in Appendix A.7.

Consequently, a natural way for improving GNN robustness is to purify an adversarial graph by eliminating the high-rank components of its spectrum.

To enhance GNN robustness, Entezari et al. (2020) and Jin et al. (2020) reconstruct a graph that only preserves the low-rank components of its spectrum to mitigate the effects of adversarial attacks, via performing truncated SVD on the adjacency matrix. Specifically, given a graph adjacency matrix $\boldsymbol{A} \in \mathbb{R}^{n \times n}$, the rank-$r$ approximated adjacency matrix can be obtained via truncated SVD: $\hat{\boldsymbol{A}} = \boldsymbol{U} \Sigma \boldsymbol{V}^T$, where $\Sigma \in \mathbb{R}^{r \times r}$ is a diagonal matrix consisting of $r$ largest singular values of $\boldsymbol{A}$. $\boldsymbol{U} \in \mathbb{R}^{n \times r}$ and $\boldsymbol{V} \in \mathbb{R}^{n \times r}$ contain the corresponding left and right singular vectors, respectively.

However, the reconstructed low-rank adjacency matrix $\hat{\boldsymbol{A}}$ has two key issues: (1) $\hat{\boldsymbol{A}}$ is typically a dense matrix with $O(n^2)$ nonzero elements, which may result in prohibitively expensive storage as well as GNN training, where $n$ represents number of nodes (Entezari et al., 2020; Jin et al., 2020). Thus, existing low-rank approximation methods are not scalable to large graphs; (2) Due to the high computational cost of SVD, $\hat{\boldsymbol{A}}$ is typically obtained by only leveraging top $r$ singular values and singular vectors, where $r$ is a relatively small number (e.g., $r = 50$). Consequently, the rank of $\hat{\boldsymbol{A}}$ is only $r = 50$, which loses too much important spectrum information and thus limits the performance of GNN training.

**Reduced-rank approximation via sparsification.** To avoid the quadratic space complexity for obtaining $\hat{\boldsymbol{A}}$, one simple solution is to sparsify $\hat{\boldsymbol{A}}$ on the fly. More concretely, instead of directly computing $\hat{\boldsymbol{A}} = \boldsymbol{U} \Sigma \boldsymbol{V}^T$, we can compute $\hat{\boldsymbol{A}}$ row by row: $\hat{\boldsymbol{A}}_{i,:} = \boldsymbol{U}_{i,:} \Sigma \boldsymbol{V}^T$. Once we obtain $\hat{\boldsymbol{A}}_{i,:}$, we can sparsify it by setting $\tilde{\boldsymbol{A}}_{i,j} = 0$ if $\hat{\boldsymbol{A}}_{i,j} < \delta$ and $\tilde{\boldsymbol{A}}_{i,j} = \hat{\boldsymbol{A}}_{i,j}$ otherwise, where $\delta$ is a hyperparameter to control the sparsity. In this way, we only need to store a dense vector $\hat{\boldsymbol{A}}_{i,:}$ at a time and the final adjacency matrix $\tilde{\boldsymbol{A}}$ is sparse. Although the space complexity is reduced to $O(m)$, where $m$ denotes the number of non-zero elements in $\tilde{\boldsymbol{A}}$, this sparsification method still has quadratic time complexity for computing $\tilde{\boldsymbol{A}}$. To tackle this issue, in this work a nearly-linear complexity algorithm for constructing a reduced-rank yet sparse matrix $\hat{\boldsymbol{A}}$ is proposed by exploiting the following connection between matrix sparsification and spectral graph embedding.

**Definition 1.** *Given the top $r$ smallest eigenvalues $\lambda_1, \lambda_2, ..., \lambda_r$ and their corresponding eigenvectors $\boldsymbol{v}_1, \boldsymbol{v}_2, ..., \boldsymbol{v}_r$ of normalized graph Laplacian matrix $\boldsymbol{L}_{norm} = \boldsymbol{I} - \boldsymbol{D}^{-\frac{1}{2}} \boldsymbol{A} \boldsymbol{D}^{-\frac{1}{2}}$, where $\boldsymbol{I}$ and $\boldsymbol{A}$ are the identity matrix and graph adjacency matrix, respectively, and $\boldsymbol{D}$ is a diagonal matrix of node degrees, the **weighted spectral embedding matrix** is defined as:*

$$\boldsymbol{V} \stackrel{\text{def}}{=} \left[ \sqrt{|1 - \lambda_1|} \boldsymbol{v}_1, ..., \sqrt{|1 - \lambda_r|} \boldsymbol{v}_r \right], \tag{1}$$

*whose $i$-th row $\boldsymbol{V}_{i,:}$ is the **weighted spectral embedding** of the corresponding $i$-th node in the graph.*

**Theorem 1.** *Given a normalized graph adjacency matrix $\boldsymbol{A}_{norm} = \boldsymbol{D}^{-\frac{1}{2}} \boldsymbol{A} \boldsymbol{D}^{-\frac{1}{2}}$ of an undirected graph, let $\hat{\boldsymbol{A}}$ be the rank-$r$ approximation of $\boldsymbol{A}_{norm}$ via truncated SVD. If the top $r$ dominant eigenvalues of $\boldsymbol{A}_{norm}$ are non-negative, then $\hat{\boldsymbol{A}}_{i,j}$ corresponds to the dot product score between the weighted spectral embeddings of node $i$ and $j$.*

Theorem 1 indicates that the reduced-rank approximation of the normalized adjacency matrix via truncated SVD captures the spectral similarity between nodes, which motivates us to leverage weighted spectral embeddings to capture graph spectral (low-rank) information, thereby identifying edges to be pruned (sparsified) from the graph. Unlike traditional spectral graph embedding methods that utilize the first $r$ Laplacian eigenpairs to construct spectral embedding matrix, we exploit the top $r$ largest singular values and their singular vectors of $\boldsymbol{A}_{norm}$ that may include both the smallest and largest Laplacian eigenpairs for capturing global and local structural information, respectively (Shuman et al., 2013). More specifically, given a graph $\mathcal{G} = (\mathcal{V}, \mathcal{E})$ and its normalized adjacency matrix $\boldsymbol{A}_{norm}$, our approach first computes the $r$ largest singular values (e.g., $r = 50$) and the corresponding singular vectors of $\boldsymbol{A}_{norm}$ leveraging efficient eigensolvers in $O(r|\mathcal{E}|)$ time to construct the weighted spectral embeddings (Baglama & Reichel, 2005); next, the embedding results are then used to construct a nearest neighbor (NN) graph, where each node will be connected to its $k$ most spectrally similar nodes. Afterwards, we compute the dot product similarity scores between adjacent nodes in the NN graph. If the similarity score is less than a threshold $\delta$, we prune the corresponding edge from the NN graph to further sparsify the graph. In this work, we exploit an approximate k-nearest neighbor algorithm for constructing the NN graph, which has $O(|\mathcal{V}| \log |\mathcal{V}|)$

complexity and thus can scale to very large graphs (Malkov & Yashunin, 2018). We say a graph is a reduced-rank graph if its adjacency matrix is obtained via the proposed reduced-rank approximation method. As a result, we can obtain a reduced-rank and sparse graph in $O(r|\mathcal{E}| + |\mathcal{V}| \log |\mathcal{V}|)$ time.

Apart from the advantage of scalability, our kNN-based reduced-rank graph also preserves much more important spectrum information than the truncated SVD-based low-rank graph. As shown by Entezari et al. (2020), adversarial attacks only perturb the top few highest singular components in the graph spectrum, while the rest of spectral information corresponds to the clean graph structure in the spatial domain. Nonetheless, Figure 5 in Appendix A.8 shows the truncated SVD-based method aggressively reduces graph rank to 50, which is two orders of magnitude smaller than the rank of input graph. In contrast, our reduced-rank method only removes the highest singular components, while retaining most of important spectral information. As a result, our reduced-rank approximation kernel leads to a significant accuracy improvement over the SVD-based low-rank approximation. More details are available in Appendices A.11.

### 3.2 Adaptive Filter Learning (Phase 2)

Most existing GNN defense models (implicitly) assume the underlying graph is homophilic, implying that nearby nodes will share similar attributes; however, this assumption is not valid for heterophilic graphs in which adjacent nodes may have dissimilar attributes (Ma et al., 2021). As a consequence, existing defense methods may not be robust against graph adversarial attacks on heterophilic graphs. To address this limitation, we adopt the concept of learning a polynomial graph filter that can adapt to graph homo/heterophily (NT et al., 2020; Chien et al., 2021), and apply it on the adversarial graphs. Specifically, given a graph and its node attribute matrix $\boldsymbol{X}$, our graph filter learning process works as follows:

$$\hat{\boldsymbol{X}} = MLP(\boldsymbol{X}) \tag{2}$$

$$\boldsymbol{Z} = softmax(\sum_{p=0}^{P} c_p \boldsymbol{F}^p \hat{\boldsymbol{X}}) \tag{3}$$

where $\boldsymbol{F}$ can be either a normalized adjacency matrix $\boldsymbol{A}_{norm} = \boldsymbol{D}^{-\frac{1}{2}} \boldsymbol{A} \boldsymbol{D}^{-\frac{1}{2}}$ or normalized Laplacian matrix $\boldsymbol{L}_{norm} = \boldsymbol{I} - \boldsymbol{A}_{norm}$, $c_0, c_1, ..., c_P$ are $P + 1$ learnable polynomial coefficients. We use the output node embedding matrix $\boldsymbol{Z}$ and node training labels to calculate training loss. The weights of MLP as well as learnable polynomial coefficients are then updated via backpropagation.

Next, we are going to show that the learnable coefficients $c_0, c_1, ..., c_P$ enable the polynomial graph filter $g_P(\boldsymbol{F}) = \sum_{p=0}^{P} c_p \boldsymbol{F}^p$ in Equation 3 to adapt to a homo/heterophilic graph.

**Lemma 1** (Chien et al. (2021)). *Assume the graph $\mathcal{G}$ is connected and $|c_p| \leq 1$ $\forall p \in \{0, 1, ..., P\}$. Let $\boldsymbol{F}$ to be the normalized adjacency matrix $\boldsymbol{F} = \boldsymbol{D}^{-\frac{1}{2}} \boldsymbol{A} \boldsymbol{D}^{-\frac{1}{2}}$. If $c_p \geq 0$ $\forall p \in \{0, 1, ..., P\}$, $\sum_{p=0}^{P} c_p = 1$, and $\exists p' > 0$ such that $c_{p'} > 0$, then $g_p(\boldsymbol{F})$ is a low-pass filter. Also, if $c_p = (-\alpha)^k, \alpha \in (0, 1)$ and P is large enough, then $g_p(\boldsymbol{F})$ is a high-pass filter.*

Intuitively, Lemma 1 indicates that for a homophilic graph the filter coefficients tend to be positive, whereas for a heterophilic graph both positive and negative coefficients can be observed. Consequently, $g_P(\boldsymbol{F})$ can be learned to serve as a low-pass (high-pass) graph filter to filter node intermediate embeddings $\hat{\boldsymbol{X}}$ over the homophilic (heterophilic) graph, such that nearby nodes will have similar (dissimilar) output embeddings. Thus, a natural idea for developing a robust GNN model on both homophilic and heterophilic graphs is to exploit both the reduced-rank graph in Section 3.1 and the aforementioned adaptive polynomial graph filter.

However, adaptive polynomial graph filters typically perform well on clean graphs that are entirely homophilic or heterophilic, while the adversarial graphs can be globally homophilic yet locally heterophilic (Zhu et al., 2021). Another important question is whether the polynomial filters can still be effective when combined with reduced-rank approximation of the adversarial graph. Next, we answer this question by showing the upper bound of the performance difference between the polynomial graph filter on a clean graph and that on the corresponding reduced-rank graph.

**Theorem 2.** *Let $\boldsymbol{A}_{clean}$, $\boldsymbol{A}_{adv}$, and $\boldsymbol{A}_r$ represent the normalized adjacency matrices of a clean graph, the corresponding adversarial graph, and the rank-$r$ approximated adversarial graph, respectively. Also let $\sigma_{r+1}$ denote the $(r + 1)$-th largest singular value of $\boldsymbol{A}_{adv}$. Assume the spectral*

*norm of graph adversarial perturbation is upper bounded by a constant $\epsilon$, i.e., $\|\boldsymbol{A}_{clean} - \boldsymbol{A}_{adv}\|_2 \leq \epsilon$. Given a polynomial graph filter $g_P(\boldsymbol{F}) = \sum_{p=0}^{P} c_p \boldsymbol{F}^p$ , we have:*

$$\|g_P(\boldsymbol{A}_{clean}) - g_P(\boldsymbol{A}_r)\|_2 \leq \sum_{p=1}^{P} p\,|c_p|\,(\epsilon + \sigma_{r+1}) \tag{4}$$

Note that the graph attacking algorithms are typically designed to perturb the graph structure in an unnoticeable way, which means that $\epsilon$ is a small value. Moreover, $\sigma_{r+1}$ is within the range of $[0, 1]$ since $\boldsymbol{A}_{adv}$ is normalized. We further set $P \leq 10$ and enforce $\sum_{p=0}^{P} |c_p| = 1$ when learning the adaptive filter to tighten the upper bound. As a result, Theorem 2 indicates that the performance of a polynomial filter on the reduced-rank graph will be similar to its performance on the clean graph. Hence, applying the adaptive graph filter on the reduced-rank graph constructed in Section 3.1 allows GARNET to work effectively for both homophilic and heterophilic graphs under adversarial attacks.

**Scalability of adaptive filter learning.** To scale the adaptive filter learning process to large graphs, we do not explicitly form the graph filter $g_P(\boldsymbol{F}) = \sum_{p=0}^{P} c_p \boldsymbol{F}^p$. Instead, we iteratively left-multiply $\hat{\boldsymbol{X}}$ by $\boldsymbol{F}$ in Equation 3 to leverage the sparsity of $\boldsymbol{F}$. Thus, the time complexity is linear to the graph size. Besides, we can effectively exploit model parallelism (Castelló et al., 2019) , where the computation of a graph filter $g_P(\boldsymbol{F})$ is distributed onto multiple GPUs based on the index $p$ that indicates the $p$-th term in $g_P(\boldsymbol{F})$. This way, we can dramatically reduce the memory usage per GPU, thereby allowing the adaptive filter learning process to scale to large graphs with millions of nodes.

### 3.3 ADAPTIVE LABEL PROPAGATION (PHASE 3)

Once the trained graph filter $g_P^*(\boldsymbol{F}) = \sum_{p=0}^{P} c_p^* \boldsymbol{F}^p$ is obtained, we further leverage it to enhance the quality of node embeddings $\boldsymbol{Z}$ in Equation 3 by correlating residue errors and propagating node labels, which is partly inspired by the correct and smooth (C&S) method recently proposed in (Huang et al., 2020). The key idea of C&S is to smooth the residue errors and node labels over the graph leveraging a low-pass graph filter, which assumes the underlying graph to be homophilic. In contrast, our approach works effectively for both homophilic and heterophilic graphs by exploiting adaptive filter learning. Specifically, let $\mathbb{N}_L$ and $\mathbb{N}_U$ be the set of nodes that are labelled and unlabelled, respectively. Given the one-hot label matrix $\boldsymbol{Y} \in \mathbb{R}^{n \times c}$, where $n$ and $c$ denote the numbers of nodes and classes, respectively, we define the residue error matrix by $\boldsymbol{R} \in \mathbb{R}^{n \times c}$ such that $\boldsymbol{R}_{\mathbb{N}_L} = \boldsymbol{Z}_{\mathbb{N}_L} - \boldsymbol{Y}_{\mathbb{N}_L}$ and $\boldsymbol{R}_{\mathbb{N}_U} = 0$. Our goal is to optimize the following objective:

$$\hat{\boldsymbol{R}} = \arg\min_{\boldsymbol{H}} \|\boldsymbol{H} - \boldsymbol{R}\|_F^2 + \lambda \operatorname{Tr}(\boldsymbol{H}^T(\boldsymbol{I} - \frac{1}{P}g_P^*(\boldsymbol{F}))\boldsymbol{H}) \tag{5}$$

where $\lambda$ is a regularization parameter. The first term in Equation 5 enforces the solution to be close to the initial residue error $\boldsymbol{R}$ that contains the label information. The key difference between our approach and the prior C&S method lies in the second term: our method spreads the error over the graph guided by the trained adaptive filter $g_P^*(\boldsymbol{F})$, whereas the C&S method adopts a simple (non-adaptive) smoothing scheme. Note that there is no training involved in this phase and $\hat{\boldsymbol{R}}$ can be iteratively computed by $\hat{\boldsymbol{R}}^{t+1} = \alpha \frac{1}{P}g_P^*(\boldsymbol{F})\hat{\boldsymbol{R}}^t + (1-\alpha)\boldsymbol{R}$ in $O(m)$ time due to the sparsity of $\boldsymbol{F}$, where $\alpha = \frac{\lambda}{1+\lambda}$ and $m$ is the number of edges in graph. After obtaining $\hat{\boldsymbol{R}}$, we update the node embeddings $\tilde{\boldsymbol{Z}}$ by $\tilde{\boldsymbol{Z}}_{\mathbb{N}_L} = \boldsymbol{Y}_{\mathbb{N}_L}$ and $\tilde{\boldsymbol{Z}}_{\mathbb{N}_U} = \boldsymbol{Z}_{\mathbb{N}_U} + \beta\hat{\boldsymbol{R}}_{\mathbb{N}_U}$, where $\beta$ is a hyperparameter. As suggested in Huang et al. (2020), we further refine $\tilde{\boldsymbol{Z}}$ by substituting $\tilde{\boldsymbol{Z}}$ for $\boldsymbol{R}$ in Equation 5 to diffuse the label information over the graph, thereby obtaining the corresponding optimizer $\hat{\boldsymbol{Z}}$. Subsequently, $\hat{\boldsymbol{Z}}$ will be utilized as the final node embeddings in the following label prediction step.

## 4 EXPERIMENTS

We have conducted comparative evaluation of GARNET against state-of-the-art defense GNN models on both homophilic and heterophilic datasets, under targeted attack (Nettack) (Zügner et al., 2018) and non-targeted attack (Mettack) (Zügner & Günnemann, 2019) with different perturbation budgets. Besides, we further evaluate the scalability of GARNET on ogbn-products, which is over $100\times$ larger than the datasets used in Entezari et al. (2020); Zhang & Zitnik (2020); Jin et al. (2020). Finally, we perform ablation studies to understand the effectiveness of each kernel in GARNET.

Table 1: Averaged node classification accuracy (%) $\pm$ std under targeted attack (Nettack) with different perturbation ratio — We denote the evaluated dataset by its name with the number of perturbations (e.g., Cora-0 means the clean Cora graph and Cora-1 denotes there is 1 adversarial edge perturbation per target node). We bold and underline the first and second highest accuracy, respectively. $OOM$ means out of memory.

| Dataset | GCN | GPRGNN | GPRSVD-CS | GCNSVD | GNNGuard | Pro-GNN | GARNET |
|---------|-----|--------|-----------|--------|----------|---------|--------|
| Cora-0 | $80.96 \pm 0.95$ | $\mathbf{84.33} \pm 2.05$ | $81.68 \pm 1.78$ | $72.65 \pm 2.29$ | $\underline{83.37} \pm 2.46$ | $81.54 \pm 1.21$ | $82.77 \pm 1.89$ |
| Cora-1 | $75.06 \pm 0.81$ | $81.68 \pm 2.18$ | $79.36 \pm 2.23$ | $70.36 \pm 1.63$ | $78.31 \pm 1.60$ | $\mathbf{82.65} \pm 0.59$ | $\underline{82.17} \pm 1.95$ |
| Cora-2 | $70.60 \pm 1.81$ | $74.34 \pm 2.41$ | $76.26 \pm 2.34$ | $65.66 \pm 2.76$ | $72.77 \pm 2.06$ | $\underline{77.83} \pm 1.10$ | $\mathbf{78.55} \pm 2.11$ |
| Cora-3 | $69.04 \pm 3.31$ | $70.96 \pm 2.00$ | $70.90 \pm 3.89$ | $61.20 \pm 1.93$ | $68.19 \pm 2.48$ | $\underline{71.08} \pm 1.20$ | $\mathbf{79.40} \pm 1.35$ |
| Cora-4 | $61.69 \pm 1.48$ | $65.90 \pm 1.61$ | $65.51 \pm 3.27$ | $57.34 \pm 3.46$ | $62.41 \pm 2.65$ | $\underline{67.83} \pm 1.87$ | $\mathbf{72.77} \pm 2.16$ |
| Cora-5 | $55.66 \pm 1.95$ | $62.89 \pm 1.95$ | $63.52 \pm 3.27$ | $55.30 \pm 2.25$ | $57.59 \pm 2.46$ | $\underline{65.38} \pm 1.65$ | $\mathbf{71.45} \pm 2.73$ |
| Citeseer-0 | $81.59 \pm 0.82$ | $82.38 \pm 0.82$ | $82.38 \pm 0.50$ | $80.95 \pm 1.23$ | $80.32 \pm 1.34$ | $\underline{82.89} \pm 1.53$ | $\mathbf{83.86} \pm 1.07$ |
| Citeseer-1 | $79.04 \pm 1.80$ | $80.15 \pm 0.84$ | $81.38 \pm 0.50$ | $75.23 \pm 2.67$ | $78.57 \pm 2.14$ | $\underline{81.74} \pm 0.79$ | $\mathbf{83.49} \pm 1.14$ |
| Citeseer-2 | $76.19 \pm 3.89$ | $80.32 \pm 0.82$ | $80.27 \pm 0.67$ | $60.15 \pm 2.29$ | $73.18 \pm 4.56$ | $\underline{80.15} \pm 0.71$ | $\mathbf{80.63} \pm 1.46$ |
| Citeseer-3 | $62.54 \pm 4.50$ | $77.46 \pm 1.46$ | $\mathbf{78.95} \pm 0.74$ | $58.89 \pm 5.28$ | $64.38 \pm 5.89$ | $\underline{78.36} \pm 2.56$ | $76.67 \pm 2.25$ |
| Citeseer-4 | $57.30 \pm 3.62$ | $73.33 \pm 3.16$ | $\underline{73.95} \pm 0.75$ | $51.74 \pm 7.96$ | $59.05 \pm 5.96$ | $\mathbf{73.98} \pm 1.28$ | $72.22 \pm 2.28$ |
| Citeseer-5 | $51.75 \pm 2.50$ | $67.89 \pm 3.74$ | $\underline{67.95} \pm 1.98$ | $45.07 \pm 2.77$ | $54.13 \pm 8.05$ | $67.46 \pm 6.36$ | $\mathbf{68.19} \pm 4.03$ |
| Pubmed-0 | $87.26 \pm 0.51$ | $90.05 \pm 0.73$ | $OOM$ | $87.03 \pm 0.48$ | $89.57 \pm 0.28$ | $OOM$ | $\mathbf{90.99} \pm 0.52$ |
| Pubmed-1 | $86.29 \pm 0.68$ | $\underline{89.30} \pm 0.54$ | $OOM$ | $84.46 \pm 0.28$ | $87.84 \pm 0.51$ | $OOM$ | $\mathbf{90.91} \pm 0.47$ |
| Pubmed-2 | $83.17 \pm 0.67$ | $\underline{87.42} \pm 0.28$ | $OOM$ | $82.68 \pm 0.46$ | $85.00 \pm 0.59$ | $OOM$ | $\mathbf{90.75} \pm 0.55$ |
| Pubmed-3 | $81.13 \pm 0.53$ | $\underline{84.46} \pm 0.53$ | $OOM$ | $81.34 \pm 0.68$ | $81.29 \pm 0.90$ | $OOM$ | $\mathbf{90.70} \pm 0.37$ |
| Pubmed-4 | $75.48 \pm 0.52$ | $81.72 \pm 0.72$ | $OOM$ | $\underline{82.41} \pm 0.54$ | $76.07 \pm 0.77$ | $OOM$ | $\mathbf{90.11} \pm 0.57$ |
| Pubmed-5 | $66.67 \pm 1.34$ | $76.99 \pm 1.16$ | $OOM$ | $\underline{79.56} \pm 0.48$ | $69.89 \pm 1.18$ | $OOM$ | $\mathbf{89.52} \pm 0.45$ |

**Experimental Setup.** The details of datasets are available in Appendix A.9.2. We evaluate unvaccinated GCN (Kipf & Welling, 2016) and GPRGNN (Chien et al., 2021). Moreover, we choose as the defense baselines three state-of-the-art vaccinated GNN models: GCNSVD (Entezari et al., 2020), GNNGuard (Zhang & Zitnik, 2020), and Pro-GNN Jin et al. (2020). Besides, we further evaluate a strong defense baseline GPRSVD-CS by combining truncated SVD (for low-rank approximation), GPRGNN (for adaptive filter learning), and C&S (for label propagation). For all baselines, we tune their hyperparameters against adversarial attacks with a small perturbation, and keep the same hyperparameters for larger adversarial perturbations. We further show hyperparameter settings of GARNET and hardware information in Appendix A.9.

## 4.1 DEFENSE ON HOMOPHILIC GRAPHS

**Defense against targeted attack.** We first evaluate the model robustness against Nettack, a strong attack method to fool a GNN model to misclassify some target nodes with a few structure (edge) perturbations. We choose the same set of target nodes as in (Jin et al., 2020). We report the averaged accuracy over 10 runs on Cora, Citeseer, and Pubmed datasets with adversarial perturbations varying from 0 to 5 per target node. Table 1 shows that GARNET outperforms all the baselines in most cases, with the accuracy improvement up to $9.96\%$ over existing defense methods. Note that the accuracy degradation of GARNET when increasing perturbation budget is much smaller than that of baselines. For instance, the accuracy drop of GARNET is only $1.39\%$ with perturbations varying from 1 to 5 on Pubmed, while the accuracy of baseline defense methods drops $4.9\% \sim 17.95\%$, indicating that GARNET is indeed more robust to strong targeted attack on homophilic graphs. Moreover, GARNET gains up to $8.5\%$ accuracy improvement over GPRSVD-CS, which reveals that our reduced-rank approximation kernel produces a much higher quality of low-rank graph than the SVD-based graph by preserving more useful spectrum information as explained in Section 3.1.

**Defense against non-targeted attack.** We further evaluate model robustness against a strong non-targeted attack, i.e., Metattack, whose goal is to drop the overall accuracy of the whole test set with a given perturbation ratio budget (i.e., the number of adversarial edges over the number of total edges). We report the averaged accuracy over 10 runs on Cora, Citeseer, and Pubmed with perturbation ratio in $\{0\%, 10\%, 20\%\}$. As shown in Table 2, GARNET consistently achieves the highest adversarial accuracy across all datasets under different attack perturbation ratios, which verifies that GARNET can also successfully defend against the non-targeted attack on homophilic graphs. It is worth mentioning that both GPRSVD-CS and Pro-GNN run out of memory even on Pubmed, a graph with only 20k nodes. In contrast, GARNET is not only robust to adversarial attacks, but also scalable to large graphs, as empirically shown in Section 4.3.

Table 2: Averaged node classification accuracy (%) $\pm$ std under non-targeted attack (Metattack) with different perturbation ratio — We denote the evaluated dataset by its name with the perturbation ratio (e.g., Cora-0 means the clean Cora graph and Cora-10 denotes there are $10\%$ adversarial edges). We bold and underline the first and second highest accuracy, respectively. $OOM$ means out of memory.

| Dataset | GCN | GPRGNN | GPRSVD-CS | GCNSVD | GNNGuard | Pro-GNN | GARNET |
|---------|-----|--------|-----------|--------|----------|---------|--------|
| Cora-0 | $83.35 \pm 0.66$ | $\underline{85.05} \pm 0.42$ | $82.61 \pm 0.54$ | $73.86 \pm 0.53$ | $84.45 \pm 0.63$ | $\mathbf{85.56} \pm 0.36$ | $82.67 \pm 1.89$ |
| Cora-10 | $69.50 \pm 1.46$ | $80.37 \pm 0.65$ | $\underline{81.08} \pm 0.52$ | $69.45 \pm 0.69$ | $69.35 \pm 2.29$ | $77.90 \pm 0.69$ | $\mathbf{82.17} \pm 0.69$ |
| Cora-20 | $56.28 \pm 1.19$ | $74.27 \pm 2.11$ | $\underline{78.50} \pm 2.20$ | $62.44 \pm 1.16$ | $64.17 \pm 2.00$ | $72.28 \pm 1.67$ | $\mathbf{81.34} \pm 0.79$ |
| Citeseer-0 | $72.15 \pm 0.75$ | $\underline{74.18} \pm 0.55$ | $73.09 \pm 0.89$ | $68.33 \pm 1.17$ | $72.09 \pm 1.09$ | $73.29 \pm 1.49$ | $\mathbf{74.82} \pm 1.07$ |
| Citeseer-10 | $67.38 \pm 1.56$ | $72.13 \pm 0.61$ | $71.75 \pm 0.85$ | $68.29 \pm 0.70$ | $67.22 \pm 2.60$ | $\underline{72.50} \pm 0.53$ | $\mathbf{74.25} \pm 0.63$ |
| Citeseer-20 | $57.21 \pm 1.26$ | $68.44 \pm 0.90$ | $62.33 \pm 1.08$ | $68.47 \pm 0.72$ | $55.30 \pm 2.23$ | $\underline{71.10} \pm 0.72$ | $\mathbf{72.03} \pm 0.50$ |
| Pubmed-0 | $\underline{87.16} \pm 0.09$ | $\mathbf{87.35} \pm 0.13$ | $OOM$ | $84.53 \pm 0.08$ | $85.38 \pm 0.17$ | $OOM$ | $86.86 \pm 0.57$ |
| Pubmed-10 | $81.16 \pm 0.13$ | $\underline{85.52} \pm 0.14$ | $OOM$ | $84.56 \pm 0.10$ | $77.45 \pm 0.20$ | $OOM$ | $\mathbf{86.24} \pm 0.20$ |
| Pubmed-20 | $77.20 \pm 0.27$ | $84.18 \pm 0.15$ | $OOM$ | $\underline{84.30} \pm 0.08$ | $71.73 \pm 0.32$ | $OOM$ | $\mathbf{85.69} \pm 0.26$ |

## 4.2 DEFENSE ON HETEROPHILIC GRAPHS

To evaluate the robustness on heterophilic graphs, we apply both targeted (Nettack) and non-targeted (Metattack) attacks on Chameleon and Squirrel datasets. Due to the space limitation, we only report the adversarial accuracy under Metattack in Table 3. The results under Nettack are available in Appendix A.4. As shown in Table 3, all defense baselines are not robust to graph adversarial attacks on heterophilic graphs. Moreover, the performance of those vaccinated models is in fact drastically worse than that of the unvaccinated model GPRGNN, a model designed to handle heterophilic graphs. Our approach, on the other hand, consistently outperforms all defense baselines by a large margin. For instance, GARNET achieves $18.06\%$ accuracy improvement over the best vaccinated model (i.e., GPRSVD-CS) on Chameleon with a $20\%$ perturbation ratio. Moreover, GARNET also consistently increases adversarial accuracy over GPRGNN. Apart from achieving the highest accuracy, the accuracy drop of GARNET is much smaller compared to that of baselines when increasing the perturbation ratio. Specifically, the accuracy of GARNET drops $1.15\%$ on Chameleon when increasing the perturbation ratio from $0\%$ to $20\%$, while the accuracy of baselines drops $2.10\% \sim 18.12\%$. Thus, GARNET is also robust against graph adversarial attacks on heterophilic graphs.

Table 3: Averaged node classification accuracy (%) $\pm$ std under non-targeted attack (Metattack) with different perturbation ratio — We denote the evaluated dataset by its name with the perturbation ratio (e.g., Chameleon-0 means the clean Chameleon graph and Chameleon-10 denotes there are $10\%$ adversarial edges). We bold and underline the first and second highest accuracy, respectively.

| Dataset | GCN | GPRGNN | GPRSVD-CS | GCNSVD | GNNGuard | Pro-GNN | GARNET |
|---------|-----|--------|-----------|--------|----------|---------|--------|
| Chameleon-0 | $58.16 \pm 1.29$ | $\mathbf{61.36} \pm 1.00$ | $47.29 \pm 1.63$ | $45.08 \pm 0.70$ | $58.01 \pm 1.57$ | $47.37 \pm 1.93$ | $\underline{61.11} \pm 2.46$ |
| Chameleon-10 | $43.47 \pm 0.78$ | $\underline{57.55} \pm 1.26$ | $47.07 \pm 1.21$ | $41.95 \pm 0.39$ | $41.75 \pm 0.93$ | $38.39 \pm 1.35$ | $\mathbf{60.96} \pm 1.22$ |
| Chameleon-20 | $39.58 \pm 1.56$ | $\underline{53.20} \pm 0.88$ | $45.12 \pm 1.34$ | $40.90 \pm 0.77$ | $39.89 \pm 1.34$ | $32.24 \pm 1.53$ | $\mathbf{59.96} \pm 0.84$ |
| Squirrel-0 | $37.45 \pm 0.76$ | $\underline{39.51} \pm 1.64$ | $31.36 \pm 1.87$ | $31.17 \pm 0.47$ | $37.46 \pm 0.56$ | $32.02 \pm 2.11$ | $\mathbf{43.43} \pm 1.14$ |
| Squirrel-10 | $26.96 \pm 0.30$ | $\underline{38.27} \pm 0.83$ | $28.25 \pm 1.66$ | $25.83 \pm 0.32$ | $27.03 \pm 0.54$ | $26.03 \pm 1.23$ | $\mathbf{42.62} \pm 1.09$ |
| Squirrel-20 | $23.94 \pm 0.45$ | $\underline{35.22} \pm 1.20$ | $23.91 \pm 1.40$ | $14.90 \pm 0.60$ | $23.69 \pm 0.59$ | $20.09 \pm 3.55$ | $\mathbf{41.97} \pm 1.02$ |

## 4.3 DEFENSE ON LARGE GRAPHS

Table 4: Averaged accuracy (%) $\pm$ std and run time under non-targeted attack (DICE) with $60\%$ perturbation ratio — We bold the highest accuracy. $OOM$ indicates out of memory.

| Method | ogbn-arxiv | | | ogbn-products | | |
|--------|-------|-------------|-------------|-------|-------------|-------------|
| | Clean | Adversarial | Time (mins) | Clean | Adversarial | Time (mins) |
| GNNGuard | $68.22 \pm 1.80$ | $60.46 \pm 2.71$ | $22.21$ | $74.82 \pm 0.11$ | $66.69 \pm 0.12$ | $567.11$ |
| GARNET | $\mathbf{70.23} \pm 0.45$ | $\mathbf{62.89} \pm 0.22$ | $1.08\ (20\times)$ | $\mathbf{81.65} \pm 0.11$ | $\mathbf{75.61} \pm 0.14$ | $31.27\ (18\times)$ |

To demonstrate the scalability of GARNET on large graphs, we evaluate the robustness of GARNET on the attacked ogbn-arxiv and ogbn-products. Given that existing strongest attacks (Nettack and Metattack) are not scalable to large graphs, we leverage a less powerful yet more scalable attacking algorithm called DICE (Waniek et al., 2018), which randomly connects (disconnects) nodes from

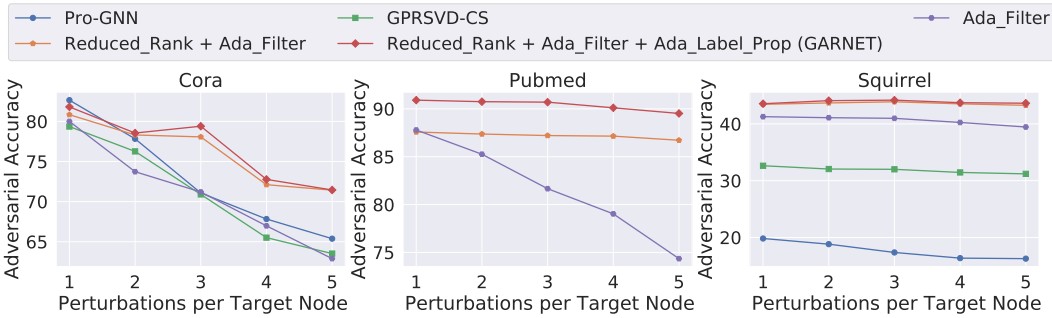

Figure 2: Comparisons of different kernel combinations in GARNET on Cora, Pubmed, and Squirrel datasets under Nettack — We denote the reduced-rank approximation kernel, adaptive filter learning kernel, and adaptive label propagation kernel by Reduced_Rank, Ada_Filter, and Ada_Label_Prop, respectively. Besides, we choose ProGNN and GPRSVD-CS (GPRGNN + SVD + C&S) as strong baselines, whose curves on Pubmed are missing due to out-of-memory.

different (same) classes, to perturb the graph structure. To have a challenging defense scenario, we consider a large perturbation budget for DICE while keeping the overall graph size the same. Specifically, we use DICE to first randomly delete $30\%$ edges linking nodes from the same class and then randomly insert $30\%$ edges linking nodes from different classes. We evaluate GNNGuard as a defense baseline, since all other defense baselines (i.e., GPRSVD-CS, GCNSVD, and Pro-GNN) run out of memory on these two datasets. Table 4 reports the accuracy and run time on large graphs under DICE attack, which shows GARNET outperforms GNNGuard by a margin of $2.43\%$ and $8.92\%$ on ogbn-arxiv and ogbn-products, respectively, with around $20\times$ runtime speedup. We further show the run time per kernel of GARNET in Appendix A.5.

## 4.4 ABALATION ANALYSIS ON GARNET KERNELS

To study the effectiveness of our proposed GARNET kernels separately, we conduct experiments by adding three kernels one by one to see how each kernel affects the adversarial accuracy. Specifically, we evaluate the model robustness against Nettack with the number of perturbations varying from 1 to 5 per target node. We show results on three datasets (two homophilic datasets Cora & Pubmed and one heterophilic dataset Squirrel) in Figure 2. When we only train an adaptive graph filter for node prediction, the adversarial accuracy is comparable to the accuracy of Pro-GNN on homophilic datasets, while it largely outperforms Pro-GNN on the heterophilic dataset, which shows the adaptive filter works effectively on both homophilic and heterophilic graphs. Figure 2 further shows that adding the reduced-rank approximation kernel results in a much more graceful accuracy degradation when increasing the budget of adversarial perturbations. For instance, the accuracy achieved by combining reduced-rank approximation and adaptive filter remains almost the same when increasing the perturbation from 1 to 5 per target node on Pubmed. This implies that our reduced-rank approximation kernel can effectively remove the high-rank adversarial properties from the input graph, allowing GARNET to be more resistant to graph adversarial attacks. It is worth noting that existing SVD-based low-rank methods (e.g., GPRSVD-CS) performs much worse than our reduced-rank method, which lies in the fact that our reduced-rank kernel preserves much more important spectrum information than SVD-based method as analyzed in Section 3.1. Furthermore, incorporating the adaptive label propagation kernel also helps improve the adversarial accuracy, implying that propagating node labels by using the adaptive filter can effectively exploit the label information for further enhancing the quality of node representations under adversarial attacks.

## 5 CONCLUSIONS

This work introduces GARNET, a spectral approach to robust and scalable graph neural networks. GARNET first construct a reduced-rank yet sparse approximation of the adversarial graph; then it trains an adaptive graph filter to obtain refined node representations as well as the learned adaptive filter that will subsequently guide the process of label propagation to further enhance the quality of node representations. Results show that GARNET outperforms state-of-the-art defense models on homophilic and heterophilic graphs under both targeted and non-targeted adversarial attacks.

## REPRODUCIBILITY STATEMENT

The GARNET source code is available at github.com/gnngarnet/garnet. Besides, we provide our proofs for Theorems 1 and 2 in Appendices A.1 and A.2, respectively. Moreover, the proof for Lemma 1 is available in Chien et al. (2021). Finally, the details of all datasets used in our experiments are available in Appendix A.9.

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

## A APPENDIX

### A.1 PROOF FOR THEOREM 1

*Proof.* As the graph is undirected, we can perform eigendecomposition on both $\boldsymbol{A}_{norm}$ and $\boldsymbol{L}_{norm}$. Let $\lambda_i$, $\hat{\lambda}_i$, and $\sigma_i$, $i = 1, 2, ..., r$ denote the $r$ smallest eigenvalues of $\boldsymbol{L}_{norm}$, $r$ largest eigenvalues of $\boldsymbol{A}_{norm}$, and $r$ largest singular values of $\boldsymbol{A}_{norm}$, respectively. Since $\boldsymbol{A}_{norm} = \boldsymbol{I} - \boldsymbol{L}_{norm}$, $\boldsymbol{A}_{norm}$ and $\boldsymbol{L}_{norm}$ share the same set of eigenvectors while their eigenvalues satisfy: $\hat{\lambda}_i = 1 - \lambda_i$, $i = 1, 2, ..., r$. Moreover, since we assume that the $r$ largest magnitude eigenvalues of $\boldsymbol{A}_{norm}$ are non-negative, we have $\sigma_i = \left|\hat{\lambda}_i\right| = \hat{\lambda}_i$, $i = 1, 2, ..., r$. Thus, we have:

$$\boldsymbol{V}\boldsymbol{V}^T = [\boldsymbol{v}_1, ..., \boldsymbol{v}_r] \begin{bmatrix} |1 - \lambda_1| & & \\ & \ddots & \\ & & |1 - \lambda_r| \end{bmatrix} [\boldsymbol{v}_1, ..., \boldsymbol{v}_r]^T$$

$$= [\boldsymbol{v}_1, ..., \boldsymbol{v}_r] \begin{bmatrix} \left|\hat{\lambda}_1\right| & & \\ & \ddots & \\ & & \left|\hat{\lambda}_r\right| \end{bmatrix} [\boldsymbol{v}_1, ..., \boldsymbol{v}_r]^T$$

$$= [\boldsymbol{v}_1, ..., \boldsymbol{v}_r] \begin{bmatrix} \sigma_1 & & \\ & \ddots & \\ & & \sigma_r \end{bmatrix} [\boldsymbol{v}_1, ..., \boldsymbol{v}_r]^T$$

$$= \hat{\boldsymbol{A}}$$

Since $\boldsymbol{V}_{i,:}$ is defined as the weighted spectral embedding of node $i$ and $\hat{\boldsymbol{A}}_{i,j} = \boldsymbol{V}_{i,:}\boldsymbol{V}_{j,:}^T$, $\hat{\boldsymbol{A}}_{i,j}$ corresponds to the dot product score between the weighted spectral embeddings of node $i$ and $j$, which completes the proof of the theorem. □

### A.2 PROOF FOR THEOREM 2

To prove Theorem 2, we first give two Lemmas below:

**Lemma 2.** *Given a graph adjacency matrix $\boldsymbol{A}$ and degree matrix $\boldsymbol{D}$, let $\boldsymbol{A}_{norm} = \boldsymbol{D}^{-\frac{1}{2}}\boldsymbol{A}\boldsymbol{D}^{-\frac{1}{2}}$, and $\hat{\boldsymbol{A}}_{norm}$ is the low-rank approximation of $\boldsymbol{A}_{norm}$ via truncated SVD, we have $\|\boldsymbol{A}_{norm}\|_2 \leq 1$ and $\|\hat{\boldsymbol{A}}_{norm}\|_2 \leq 1$*

The proof for Lamma 2 is trivial since the spectrum of $\boldsymbol{A}_{norm}$ and $\hat{\boldsymbol{A}}_{norm}$ lies in $[-1, 1]$ (Chung, 1997).

**Lemma 3.** *Given two matricies $\boldsymbol{M}$ and $\boldsymbol{N}$ such that $\|\boldsymbol{M}\|_2 \leq 1$ and $\|\boldsymbol{N}\|_2 \leq 1$, we have $\|\boldsymbol{M}^p - \boldsymbol{N}^p\|_2 \leq p\|\boldsymbol{M} - \boldsymbol{N}\|_2$ for every $p \geq 0$.*

The proof for Lamma 3 is available in Levie et al. (2019).

Next, we formally prove Theorem 2 in the following:

*Proof.*

$$\|g_P(\boldsymbol{A}_{clean}) - g_P(\boldsymbol{A}_r)\|_2 = \|\sum_{p=0}^{P} c_p \boldsymbol{A}_{clean}^p - \sum_{p=0}^{P} c_p \boldsymbol{A}_r^p\|_2$$

$$= \sum_{p=1}^{P} \|c_p(\boldsymbol{A}_{clean}^p - \boldsymbol{A}_r^p)\|_2$$

$$\leq \sum_{p=1}^{P} |c_p| \|\boldsymbol{A}_{clean}^p - \boldsymbol{A}_r^p\|_2$$

$$\leq \sum_{p=1}^{P} p \, |c_p| \, \|\boldsymbol{A}_{clean} - \boldsymbol{A}_r\|_2$$

$$\leq \sum_{p=1}^{P} p \, |c_p| \, (\|\boldsymbol{A}_{clean} - \boldsymbol{A}_{adv}\|_2 + \|\boldsymbol{A}_{adv} - \boldsymbol{A}_r\|_2)$$

$$\leq \sum_{p=1}^{P} p \, |c_p| \, (\epsilon + \sigma_{r+1})$$

The second inequality above is based on Lemmas 2 and 3, while the third inequality is derived by using the triangle inequality. □

## A.3 GARNET Algorithm and Complexity Analysis

---

**Algorithm 1:** GARNET algorithm

---

**Input:** Adjacency matrix $\boldsymbol{A} \in \mathbb{R}^{n \times n}$; node feature matrix $\boldsymbol{X} \in \mathbb{R}^{n \times d}$; node label matrix $\boldsymbol{Y} \in \mathbb{R}^{n \times c}$; truncated svd rank $r$; kNN graph $k$; polynomial filter degree $P$; error correction scale $\beta$; train node set $\mathbb{N}_L$; test node set $\mathbb{N}_U$; label propagation iteration $s$; regularization parameter $\alpha$

**Output:** Node embedding matrix $\hat{\boldsymbol{Z}} \in \mathbb{R}^{n \times c}$

/* Phase1:reduced-rank approximation                    */

1   $\boldsymbol{U}, \boldsymbol{S}, \boldsymbol{V}^T = truncated\_svd(\boldsymbol{A}, rank = r)$;

2   $\tilde{\boldsymbol{A}} = kNN\_graph(\boldsymbol{U}\sqrt{\boldsymbol{S}}, k)$;

/* Phase2:adaptive filter learning                      */

3   **for** $1...epochs$ **do**

4     $\hat{\boldsymbol{X}} = MLP(\boldsymbol{X})$;

5     $\boldsymbol{Z} = Softmax(\sum_{p=0}^{P} c_p \tilde{\boldsymbol{A}}^p \hat{\boldsymbol{X}})$;

6     $loss = CrossEntropyLoss(\boldsymbol{Z}, \boldsymbol{Y})$;

7     $loss.backward()$;

8   **end**

/* Phase3:adaptive label propagation                    */

9   $\boldsymbol{R}_{\mathbb{N}_L} = \boldsymbol{Z}_{\mathbb{N}_L} - \boldsymbol{Y}_{\mathbb{N}_L}$;

10   $\boldsymbol{R}_{\mathbb{N}_U} = 0$;

11   $\hat{\boldsymbol{R}}^0 = \boldsymbol{R}$;

12   **for** $i = 0...s$ **do**

13     $\hat{\boldsymbol{R}}^{i+1} = \frac{\alpha}{P} \sum_{p=0}^{P} c_p^* \tilde{\boldsymbol{A}}^p \hat{\boldsymbol{R}}^i + (1-\alpha)\boldsymbol{R}$;

14   **end**

15   $\tilde{\boldsymbol{Z}}_{\mathbb{N}_L} = \boldsymbol{Y}_{\mathbb{N}_L}$;

16   $\tilde{\boldsymbol{Z}}_{\mathbb{N}_U} = \boldsymbol{Z}_{\mathbb{N}_U} + \beta \hat{\boldsymbol{R}}_{\mathbb{N}_U}$;

17   $\hat{\boldsymbol{Z}}^0 = \tilde{\boldsymbol{Z}}$;

18   **for** $i = 0...s$ **do**

19     $\hat{\boldsymbol{Z}}^{i+1} = \frac{\alpha}{P} \sum_{p=0}^{P} c_p^* \tilde{\boldsymbol{A}}^p \hat{\boldsymbol{Z}}^i + (1-\alpha)\tilde{\boldsymbol{Z}}$;

20   **end**

---

The algorithm of GARNET is shown in Algorithm 1. We further analyze the total complexity of GARNET on a graph $\mathcal{G} = (\mathcal{V}, \mathcal{E})$. Specifically, as we mention in Section 3.1, the complexity of the truncated SVD with rank $r$ and approximate kNN graph construction is $O(r|\mathcal{E}|)$ and $O(|\mathcal{V}| \log |\mathcal{V}|)$, respectively. Thus, the complexity of the reduced-rank approximation kernel is $O(r|\mathcal{E}| + |\mathcal{V}| \log |\mathcal{V}|)$. Moreover, the complexity of computing adaptive filter is $O(c|\mathcal{E}|)$, where $c$ is the column dimension of $\hat{\boldsymbol{X}}$, since we can leverage the sparsity of the graph for computing $\tilde{\boldsymbol{A}}\hat{\boldsymbol{X}}$. Similarly, the complexity of adaptive label propagation kernel is $O(sP|\mathcal{E}|)$, where $s$ is the number of steps to iteratively compute $\hat{\boldsymbol{R}}$ and $\hat{\boldsymbol{Z}}$, and $P$ denotes the polynomial degree. Consequently, the overall complexity of GARNET is $O((r + c + sP)|\mathcal{E}| + |\mathcal{V}| \log |\mathcal{V}|)$. For the space complexity of GARNET, the reduced-rank kernel involves forming a sparse kNN graph by building hierarchical navigable small world (HNSW) graphs that contain $O(|\mathcal{V}| \log |\mathcal{V}|)$ nodes in total and each node connects to a fixed number of neighbors. Thus, it has $O(|\mathcal{V}| \log |\mathcal{V}| + |\mathcal{E}|)$ space complexity, where $|\mathcal{V}| \log |\mathcal{V}|$ represents the space usage of storing the HNSW graphs and $|\mathcal{E}|$ denotes the space usage of the constructed kNN graph. In regard to the adaptive filter learning and label propagation kernels, we do not explicitly form the graph filter $g_P(\boldsymbol{F}) = \sum_{p=0}^{P} c_p \boldsymbol{F}^p$. Instead, we iteratively left-multiply node embedding matrix $\hat{\boldsymbol{X}}$ and $\boldsymbol{H}$ by $\boldsymbol{F}$ in Equations 3 and 5, respectively, to leverage the sparsity of $\boldsymbol{F}$. Since the nonzero elements in F correspond to edges in the reduced-rank (kNN) graph and the node embedding matrix is in the shape of $|\mathcal{V}| \times d$, both adaptive filter learning and label propagation kernels have $O(|\mathcal{E}| + d|\mathcal{V}|)$ space complexity. As a result, the overall space complexity of GARNET is $O(|\mathcal{E}| + (d + \log |\mathcal{V}|)|\mathcal{V}|)$.

## A.4 ACCURACY ON HETEROPHILIC DATASETS UNDER NETTACK

Table 5: Averaged node classification accuracy (%) $\pm$ std under targeted attack (Nettack) with different perturbation ratio — We denote the evaluated dataset by its name with the number of perturbations (e.g., Chameleon-0 means the clean Chameleon graph and Chameleon-1 denotes there is 1 adversarial edge perturbation per target node). We bold and underline the first and second highest accuracy, respectively.

| Dataset | GCN | GPRGNN | GPRSVD-CS | GCNSVD | GNNGuard | Pro-GNN | GARNET |
|---|---|---|---|---|---|---|---|
| Chameleon-0 | $70.66 \pm 1.65$ | $71.46 \pm 1.92$ | $62.12 \pm 3.04$ | $68.29 \pm 0.54$ | $\mathbf{74.15} \pm 1.26$ | $60.66 \pm 3.11$ | $\underline{72.89} \pm 2.65$ |
| Chameleon-1 | $69.81 \pm 1.34$ | $71.02 \pm 1.57$ | $61.34 \pm 2.93$ | $67.93 \pm 0.56$ | $\underline{71.82} \pm 1.86$ | $59.44 \pm 3.13$ | $\mathbf{72.68} \pm 1.89$ |
| Chameleon-2 | $68.51 \pm 2.37$ | $\underline{70.71} \pm 1.12$ | $61.09 \pm 2.80$ | $68.29 \pm 0.77$ | $67.32 \pm 2.36$ | $56.44 \pm 3.13$ | $\mathbf{72.20} \pm 2.31$ |
| Chameleon-3 | $65.73 \pm 2.78$ | $\underline{70.30} \pm 1.28$ | $60.98 \pm 2.82$ | $69.15 \pm 0.95$ | $66.22 \pm 2.37$ | $52.56 \pm 3.28$ | $\mathbf{72.17} \pm 2.07$ |
| Chameleon-4 | $63.04 \pm 3.98$ | $69.87 \pm 1.29$ | $60.85 \pm 3.31$ | $\underline{70.24} \pm 0.60$ | $65.73 \pm 2.72$ | $51.95 \pm 2.59$ | $\mathbf{72.06} \pm 2.94$ |
| Chameleon-5 | $57.92 \pm 3.69$ | $66.26 \pm 1.71$ | $60.37 \pm 2.86$ | $\underline{67.44} \pm 0.78$ | $63.42 \pm 3.45$ | $51.10 \pm 2.58$ | $\mathbf{71.83} \pm 2.11$ |
| Squirrel-0 | $25.46 \pm 1.96$ | $\underline{41.36} \pm 2.87$ | $32.98 \pm 2.36$ | $31.73 \pm 1.18$ | $26.09 \pm 2.35$ | $20.45 \pm 4.52$ | $\mathbf{44.91} \pm 1.53$ |
| Squirrel-1 | $25.09 \pm 2.97$ | $\underline{41.27} \pm 3.16$ | $32.63 \pm 0.87$ | $31.00 \pm 1.11$ | $23.46 \pm 2.53$ | $19.82 \pm 4.23$ | $\mathbf{43.55} \pm 1.79$ |
| Squirrel-2 | $25.00 \pm 2.20$ | $\underline{41.09} \pm 2.14$ | $32.05 \pm 1.05$ | $30.99 \pm 1.03$ | $22.09 \pm 1.36$ | $18.82 \pm 4.17$ | $\mathbf{44.09} \pm 2.35$ |
| Squirrel-3 | $24.73 \pm 1.76$ | $\underline{40.98} \pm 2.72$ | $32.00 \pm 1.66$ | $30.18 \pm 1.67$ | $22.00 \pm 1.36$ | $17.36 \pm 4.06$ | $\mathbf{44.18} \pm 2.26$ |
| Squirrel-4 | $24.09 \pm 1.15$ | $\underline{40.25} \pm 2.82$ | $31.45 \pm 1.38$ | $30.00 \pm 0.91$ | $21.18 \pm 1.91$ | $16.36 \pm 4.16$ | $\mathbf{43.73} \pm 1.62$ |
| Squirrel-5 | $23.72 \pm 1.09$ | $\underline{39.45} \pm 2.36$ | $31.20 \pm 1.84$ | $29.54 \pm 2.51$ | $21.09 \pm 1.86$ | $16.27 \pm 3.78$ | $\mathbf{43.64} \pm 1.53$ |

As shown in Table 5, the four vaccinated baseline models, i.e., GPRSVD-CS, GCNSVD, GN-NGuard, and Pro-GNN, only achieve comparable or even worse accuracy than the unvaccinated model GPRGNN in most cases, which indicates that those defense models lose its robustness on heterophilic graphs. In contrast, GARNET largely outperforms all baselines in most scenarios. For instance, the accuracy of GARNET is 12.18% higher than the accuracy of the best baseline (i.e., GPRSVD-CS) on Squirrel with 3 perturbations per target node. Moreover, the accuracy drop of GARNET when increasing the number of perturbations per target node is only 1.06% and 1.27% on Chameleon and Squirrel, respectively, which indicates that GARNET is also resistant to the targeted attack on heterophilic graphs.

## A.5 RUN TIME PER KERNEL IN GARNET

Table 6 reports the run time of each sub-kernel in GARNET on Cora, Pubmed, ogbn-arxiv, and ogbn-products. It shows that training the adaptive graph filter is the most time consuming kernel in GARNET. It is also worth mentioning that the run time of truncated SVD and approximate kNN

Table 6: Run time (secs) of each kernel in GARNET— We decompose the reduced-rank approximation kernel into truncated SVD (TSVD) and approximate kNN graph construction (AKNN) steps. In addition, we denote the adaptive filter learning kernel and adaptive label propagation kernel by Ada_Filter, and Ada_Label_Prop, respectively.

| Dataset | TSVD | AKNN | Ada_Filter | Ada_Label_Prop | Total |
|---|---|---|---|---|---|
| Cora | 0.09 | 0.10 | 0.76 | 0.05 | 1.00 |
| Pubmed | 0.25 | 0.72 | 1.60 | 0.05 | 2.62 |
| ogbn-arxiv | 6.04 | 17.92 | 40.49 | 0.23 | 64.68 |
| ogbn-products | 368.37 | 363.32 | 1139.46 | 5.21 | 1876.36 |

graph construction is less than 7 minutes on ogbn-products, which contains 2 million of nodes and 60 million edges. This is consistent with their near-linear complexity analyzed in Section 3.1.

## A.6 ABLATION STUDY ON kNN GRAPH CONSTRUCTION

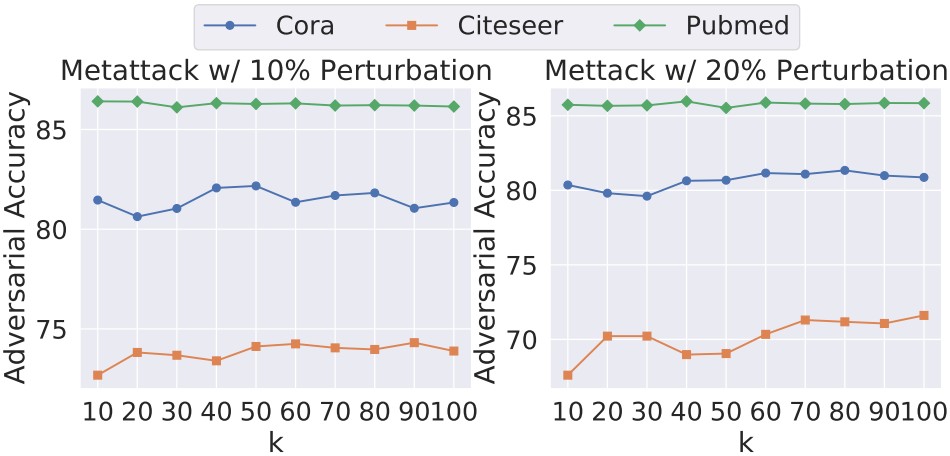

Figure 3: Ablation Study on kNN Graph.

To evaluate the sensitivity of GARNET to approximate kNN (AKNN) graph construction, we show the adversarial accuracy of GARNET with different $k$ values for constructing AKNN graphs in Figure 3, which indicates that the accuracy of GARNET does not change too much when varying $k$ value within the range of $[30, 100]$, especially on the Cora and Pubmed datasets. Hence, we recommend choosing $k = 50 \sim 80$ for the AKNN graph construction.

## A.7 GRAPH RANK UNDER ADVERSARIAL ATTACKS

To verify that applying adversarial attacks on a graph indeed increases its rank, we evaluate how the graph rank changes when increasing the perturbation ratio of Metattack. Specifically, we evaluate the graph rank growth rate on three homophilic graphs Cora, Citeseer, and Pubmed, as well as one heterophilic graph Chameleon. Figure 4 shows that the graph ranks grow with increasing perturbation ratios ($0\%$ to $25\%$). For the results showing that other graph adversarial attacks increase graph ranks, we refer to Figure 3 in Entezari et al. (2020) and Figure 1 in Jin et al. (2020).

## A.8 GRAPH RANK COMPARISON

We compute the rank of each adversarial graph under Metattack with the perturbation ratio varying from $5\%$ to $25\%$, and compare the result with its corresponding two low-rank graphs obtained via our reduced-rank approximation and truncated SVD (TSVD) respectively. Note that we use the

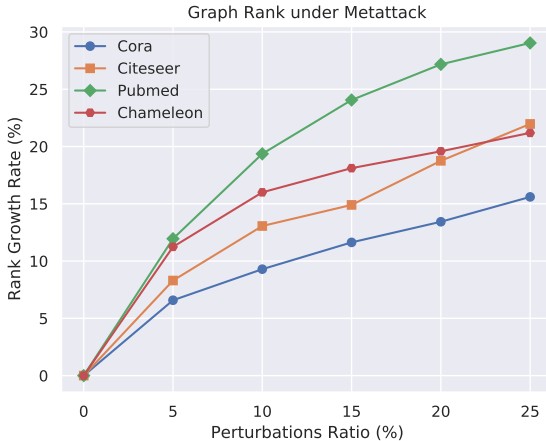

Figure 4: Graph rank growth under Metattack.

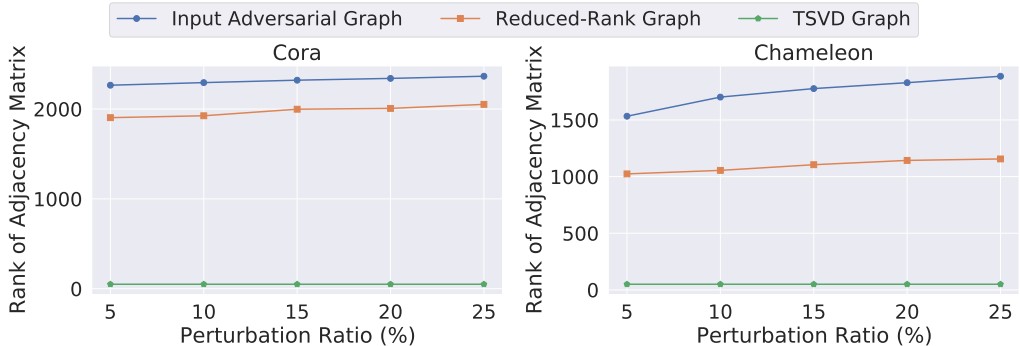

Figure 5: Comparisons of the ranks on input adversarial graph, our reduced-rank graph, and TSVD-based low-rank graph on Cora and Chameleon datasets.

same $r = 50$ largest singular values and their corresponding singular vectors for our reduced-rank kernel and TSVD. Figure 5 shows that the TSVD-based method aggressively reduces the graph rank to 50 and thus loses lots of useful spectral information. In contrast, our reduced-rank graph only removes high-rank adversarial graph properties while keeping rest of meaningful spectral information, demonstrating the effectiveness of the proposed reduced-rank approximation scheme.

## A.9 EXPERIMENTAL SETUP

### A.9.1 HYPERPARAMETER SETTINGS OF GARNET

We show the hyperparameters of the reduced-rank approximation kernel and the adaptive filter learning kernel on different datasets under Nettack (1 perturbation per node), Metattack ($10\%$ perturbation ratio), and DICE ($60\%$ perturbation ratio) in Table 7. For the adaptive label propagation kernel, we do not manually tune hyperparameters but instead leverage Optuna (Akiba et al., 2019) to optimize the hyperparameters such as the scale $\beta$ for error correction based on the validation set, since this kernel is fast to compute. We refer to github.com/gnngarnet/garnet for details of hyperparameters used in the adaptive label propagation kernel. We run all GNN training with full batch.

### A.9.2 DATASET DETAILS

Table 8 shows the statistics of the datasets used in our experiments. We follow Zhu et al. (2020) to compute the homophily score per dataset (lower score means more heterophilic). As in Jin et al.

Table 7: Summary of hyperparameters in GARNET— Apart from the notations $r$, $k$, and $\delta$ mentioned in Section 3.1, we denote polynomial degree, learning rate, number of epochs, weight_decay, dropout, and coefficient initialization in the adaptive filter learning kernel by $P$, $lr$, $epochs$, $wd$, $dp$, and $c$, respectively.

| | **Reduced-Rank Approximation** | | | **Adaptive Filter Learning** | | | | | |
| **Dataset** | $r$ | $k$ | $\delta$ | $P$ | $lr$ | $epochs$ | $wd$ | $dp$ | $c$ |
|---|---|---|---|---|---|---|---|---|---|
| Cora-Nettack | 50 | 50 | 0.05 | 10 | 0.01 | 50 | 0.0005 | 0.5 | 0.9 |
| Cora-Metattack | 50 | 50 | 0.05 | 10 | 0.01 | 50 | 0.0005 | 0.5 | 0.9 |
| Citeseer-Nettack | 50 | 30 | 0.003 | 10 | 0.01 | 50 | 0.0005 | 0.5 | 0.9 |
| Citeseer-Metattack | 50 | 60 | 0.05 | 10 | 0.01 | 50 | 0.0005 | 0.5 | 0.9 |
| Pubmed-Nettack | 50 | 80 | 0.05 | 10 | 0.01 | 150 | 0.0005 | 0.5 | 0.9 |
| Pubmed-Metattack | 50 | 80 | 0.05 | 10 | 0.01 | 150 | 0.0005 | 0.5 | 0.9 |
| Chameleon-Nettack | 50 | 60 | 0.003 | 10 | 0.05 | 300 | 0.0 | 0.5 | 0.9 |
| Chameleon-Metattack | 50 | 60 | 0.003 | 10 | 0.05 | 300 | 0.0 | 0.5 | 0.9 |
| Squirrel-Nettack | 50 | 50 | 0.003 | 10 | 0.1 | 300 | 0.0 | 0.5 | 0.3 |
| Squirrel-Metattack | 50 | 50 | 0.003 | 10 | 0.1 | 300 | 0.0 | 0.5 | 0.3 |
| ogbn-arxiv-DICE | 50 | 50 | 0.00003 | 10 | 0.01 | 500 | 0.0 | 0.0 | 0.9 |
| ogbn-products-DICE | 50 | 50 | 0.00003 | 5 | 0.01 | 300 | 0.0 | 0.0 | 0.9 |

Table 8: Statistics of datasets used in our experiments.

| Dataset | Type | Homophily Score | Nodes | Edges | Classes | Features |
|---|---|---|---|---|---|---|
| Cora | Homophily | 0.80 | $2,485$ | $5,069$ | 7 | $1,433$ |
| Citeseer | Homophily | 0.74 | $2,110$ | $3,668$ | 6 | $3,703$ |
| Pubmed | Homophily | 0.80 | $19,717$ | $44,324$ | 3 | $500$ |
| Chameleon | Heterophily | 0.23 | $2,277$ | $62,792$ | 5 | $2,325$ |
| Squirrel | Heterophily | 0.22 | $5,201$ | $396,846$ | 5 | $2,089$ |
| ogbn-arxiv | Homophily | 0.66 | $169,343$ | $1,166,243$ | 40 | $128$ |
| ogbn-products | Homophily | 0.81 | $2,449,029$ | $61,859,140$ | 47 | $100$ |

(2020), we extract the largest connected components of the original Cora, Citeseer, and Pubmed datasets (Yang et al., 2016) for the adversarial evaluation, with the same train/validation/test split. For Chameleon and Squirrel (Rozemberczki et al., 2021), we keep the same split setting as Chien et al. (2021). Finally, we follow the split setting of Open Graph Benchmark (OGB) (Hu et al., 2020) on ogbn-arxiv and ogbn-products.

In addition, we follow Jin et al. (2020) for the selection of target nodes on Cora, Citeseer, and Pubmed under Nettack. For the Chameleon and Squirrel datasets under Nettack, we choose target nodes that have degrees within the range of $[20, 50]$ and $[20, 140]$, respectively. In regard to non-targeted attacks (i.e., Metattack and DICE), we choose nodes in the test set as target nodes for all datasets.

### A.9.3 HARDWARE INFORMATION

For ogbn-arxiv and ogbn-products, we run experiments on a Linux machine with an Intel Xeon Silver 4214 CPU (8 cores @ 2.20GHz) and 4 NVIDIA RTX A6000 GPUs (48 GB memory per GPU). For the rest of the datasets, we conduct all experiments on a Linux machine with an Intel Xeon Gold 5218 CPU (8 cores @ 2.30GHz) CPU and an NVIDIA RTX 2080 Ti GPU (11 GB memory per GPU).

### A.10 ABLATION STUDY ON SPECTRAL EMBEDDING

To show that the spectral embedding used in our reduced-rank approximation kernel not only contributes to low-rank approximation but also captures key structural information per node, we compare our spectral embedding method, which uses top 50 largest singular values and corresponding singular vectors of adjacency matrix, with the vanilla spectral embedding method that leverages the top 50 smallest eigenvalues and eigenvectors of Laplacian matrix. As shown in Figure 6, the singu-

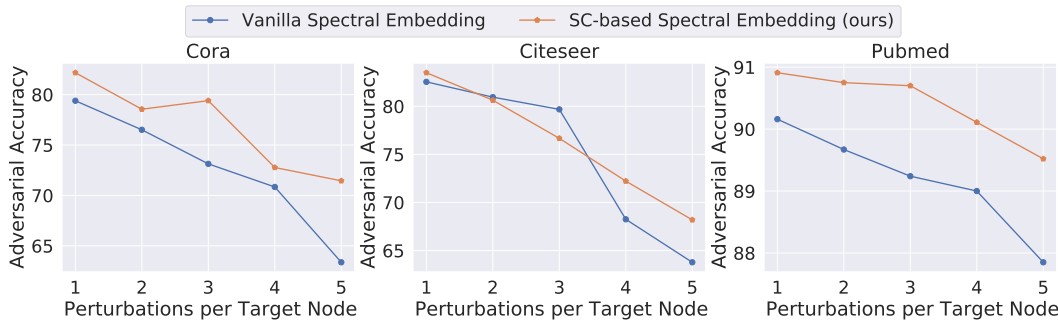

Figure 6: GARNET accuracy comparisons of using vanilla spectral embedding and singular components (SC) based spectral embedding in the reduced-rank approximation kernel.

lar component (SC) based spectral embedding outperforms the vanilla spectral embedding in most cases. The underlying reason is that the largest singular components of adjacency matrix correspond to both smallest and largest eigenvalues and their corresponding eigenvectors of Laplacian matrix, which capture global and local information, respectively (Shuman et al., 2013).

## A.11    REDUCED-RANK APPROXIMATION VS. TRUNCATED SVD

---

**Algorithm 2:** Reduced-rank approximation algorithm (Ours)

**Input:** Adjacency matrix $\boldsymbol{A} \in \mathbb{R}^{n \times n}$; kNN graph $k$; sparsification threshold $\delta$

**Output:** Reduced-rank adjacency matrix $\tilde{\boldsymbol{A}} \in \mathbb{R}^{n \times n}$

    `/* Obtain singular components via TSVD          */`

21  $\boldsymbol{U}, \boldsymbol{S}, \boldsymbol{V}^T = truncated\_svd(\boldsymbol{A}, rank = 50)$;

    `/* Node spectral embedding                      */`

22  $\boldsymbol{X} = \boldsymbol{U}\sqrt{\boldsymbol{S}}$;

    `/* Approximate kNN graph construction           */`

23  $\hat{\boldsymbol{A}} = kNN\_graph(\boldsymbol{X}, k)$;

    `/* Drop edges with node similarity scores lower`
       `than δ                                        */`

24  $\tilde{\boldsymbol{A}} = sparsification(\hat{\boldsymbol{A}}, \delta)$;

---

**Algorithm 3:** Truncated SVD based low-rank approximation algorithm

**Input:** Adjacency matrix $\boldsymbol{A} \in \mathbb{R}^{n \times n}$

**Output:** Reduced-rank adjacency matrix $\tilde{\boldsymbol{A}} \in \mathbb{R}^{n \times n}$

25  $\boldsymbol{U}, \boldsymbol{S}, \boldsymbol{V}^T = truncated\_svd(\boldsymbol{A}, rank = 50)$;

26  $\tilde{\boldsymbol{A}} = \boldsymbol{U}\boldsymbol{S}\boldsymbol{V}^T$;

---

Algorithms 2 and 3 show the difference between our reduced-rank approximation (RRA) method and truncated SVD (TSVD) low-rank approximation method. Note that the low-rank adjacency matrix $\tilde{\boldsymbol{A}}$ produced by TSVD method has two issues: (1) $\tilde{\boldsymbol{A}}$ is typically a dense matrix corresponding to a (nearly) complete graph, which cannot scale to large graphs; (2) $\tilde{\boldsymbol{A}}$ is an extremely low-rank matrix (e.g., $rank = 50$) that is two orders of magnitude smaller than the rank of input graph, as shown in Figure 5, which loses too much important spectral information.

In contrast, our RRA algorithm only involves a sparse adjacency matrix, whose density is roughly $\frac{k}{n}$ with $k \ll n$, where $k$ is the number of neighbors per node in kNN graph and $n$ is the number of

nodes in the graph. Moreover, the RRA algorithm produces a graph that only removes the highest singular components and preserves most of the important spectral information, as empirically shown in Figure 5.

Table 9: Averaged node classification accuracy (%) $\pm$ std under targeted attack (Nettack) with different perturbation ratio — We denote the evaluated dataset by its name with the number of perturbations (e.g., Cora-1 denotes there is 1 adversarial edge perturbation per target node).

| Dataset | GCN-SVD | GCN-RRA |
|---------|---------|---------|
| Cora-1 | $70.36 \pm 1.63$ | $\mathbf{79.75} \pm 2.35$ |
| Cora-2 | $65.66 \pm 2.76$ | $\mathbf{79.69} \pm 1.50$ |
| Cora-3 | $61.20 \pm 1.93$ | $\mathbf{74.42} \pm 2.06$ |
| Cora-4 | $57.34 \pm 3.46$ | $\mathbf{60.60} \pm 2.67$ |
| Cora-5 | $55.30 \pm 2.25$ | $\mathbf{59.04} \pm 2.05$ |
| Citeseer-1 | $75.23 \pm 2.67$ | $\mathbf{77.30} \pm 2.80$ |
| Citeseer-2 | $60.15 \pm 2.29$ | $\mathbf{75.23} \pm 2.14$ |
| Citeseer-3 | $58.89 \pm 5.28$ | $\mathbf{59.84} \pm 3.43$ |
| Citeseer-4 | $51.74 \pm 7.96$ | $\mathbf{57.94} \pm 5.66$ |
| Citeseer-5 | $45.07 \pm 2.77$ | $\mathbf{53.18} \pm 3.61$ |
| Pubmed-1 | $84.46 \pm 0.28$ | $\mathbf{89.03} \pm 0.68$ |
| Pubmed-2 | $82.68 \pm 0.46$ | $\mathbf{88.92} \pm 0.45$ |
| Pubmed-3 | $81.34 \pm 0.68$ | $\mathbf{88.50} \pm 0.45$ |
| Pubmed-4 | $82.41 \pm 0.54$ | $\mathbf{88.44} \pm 0.64$ |
| Pubmed-5 | $79.56 \pm 0.48$ | $\mathbf{88.12} \pm 0.86$ |

To further confirm that our RRA method produces a graph with much higher quality than the low-rank graph generated by truncated SVD, we compare the accuracy of GCN-SVD with that of GCN-RRA. As shown in Table 9, GCN-RRA consistently outperforms GCN-SVD with accuracy improvement up to $15.08\%$, which indicates that the important spectral information that RRA preserves indeed largely improves the adversarial accuracy.

