# OpenReview forum: "GARNET: A Spectral Approach to Robust and Scalable Graph Neural Networks"
_ICLR.cc/2022/Conference — ICLR 2022 Submitted_

### Official Review · Reviewer_EL8y · 2021-11-02

**Correctness:** 3
**Technical Novelty And Significance:** 3
**Empirical Novelty And Significance:** 4
**Recommendation:** 6
**Confidence:** 4

**Main Review:**

Strengths:
- The paper is easy to follow with clear motivation and is well written. The design of GARNET is clearly motivated by handling the low-homophily graphs.
- Due to the common assumption on the homophily of underlying graphs, it is of great importance to go beyond these datasets and propose a universal solution to different types of graphs.
- The acceleration of TSVD is remarkable and the scalability of the proposed method is clearly a strong contribution to the research on the robustness of GNNs.

Weaknesses:
- The main concern of mine is the contribution towards scalability is a little bit over-claimed. In the major advantages of GARNET, the authors claim that GARNET has a nearly-linear runtime/space complexity. However, the total space complexity of GARNET is not analyzed, which could result in a dense matrix after the adaptive filter learning then is not nearly linear anymore.
- Another concern is that the proposed approach is heuristic and can be easily broken by an adaptive attacker (see e.g. [1] for a detailed discussion). An adaptive attacker can easily circumvent them by changing their attack to account for the defense. For example, it is relatively straightforward to add an additional term in the computation of Nettack's scores that discourages adversarial edges that have a large impact on the high-rank spectrum. Furthermore, if the attacker knows what proposed defense is used, they can specifically target the largest singular values to potentially cause even more damage.
- For experiments, the experimental setup should be more specific for better reproducibility since no code is available for examination. Several questions include: what package do the authors use to conduct truncated SVD? I have experience in having the top eigen-pairs of the whole OGB graph with scipy, which takes hours to finish. Thus the run time that is less than 7 minutes is interesting to me. For the training on ogbn-arxiv and ogbn-products, do the authors use sampling techniques or just training with full batch? The difference of evaluation between small graphs and OGB graphs should also be specified in the detail of settings.

[1] On adaptive attacks to adversarial example defenses, NeurIPS 2020


**Summary Of The Paper:**

This paper proposes a spectral approach towards robust and scalable graph learning, namely GARNET. There are three main components inspired by handling low-homophily graphs in GARNET: low rank and sparse approximation of the adversarial graph, a variant of spectral GNN with an adaptive filter, and label propagation with special interest to the learned filter. GARNET is claimed to be nearly-linear so that it can be scaled to very huge graphs.
Experiments against both targeted and untargeted attacks under the poison setting over a collection of graph benchmark datasets demonstrate that GARNET could increase the adversarial accuracy more than state-of-the-art defending methods.


**Summary Of The Review:**

My concerns are mainly from three aspects:
- The complexity analysis is vague while it is one of the main contributions of this work.
- No adaptive attacker is evaluated.
- The reproducibility of empirical evaluation is a little weak.

---

> ### Author Response · Authors · 2021-11-16
> **Additional Clarifications**
>
> Thanks for your detailed review. We submitted the revised paper according to your comments. We are going to address your concerns one by one as follows:
>
> C1: The overall (space) complexity of GARNET.
>
> A1: As stated at the end of Section 3.2, we do not explicitly compute the adaptive filter, instead we iteratively left-multiply $\hat{X}$ by $F$ in Equation (3) to leverage the sparsity of $F$, which does not involve storing any dense adaptive filter matrix. Thus, we only need to store the sparse matrix $F$ and a node embedding matrix for both adaptive filter learning and label propagation kernels, which has $O(|E|+d|V|)$ space complexity (note that the nonzero elements in $F$ correspond to edges in our graph) . In addition, our reduced-rank kernel involves forming a sparse kNN graph by building hierarchical navigable small world (HNSW) graphs that contain $O(|V|\log|V|)$ nodes in total and each node connects to a fixed number of neighbors.  As a result, the total space complexity of GARNET is $O(|E|+(d+\log|V|)|V|)$, where $|E|$ and $|V|$ represent the number of edges and nodes in graph, respectively, and $d$ denotes the dimension of node embedding vector. We also empirically confirm this by showing results on the ogbn-products dataset, which contains millions of nodes (node that a dense adaptive filter matrix on ogbn-products can easily run out of memory on our hardware even with model parallelism). We further provided more details of time/space complexity in Appendix A.3.
>
> C2: An adaptive attack may break the robustness of GARNET.
>
> A2: We believe it is not easy to find edges that have a large impact on the high-rank spectrum, especially in an efficient and scalable way. As a result, it is actually nontrivial to enhance Nettack (or other existing attacks) by adding an additional term to discourage those specific edges. To our knowledge, all the attacks used in our experiments are strong and reasonable for evaluating model robustness.
>
> C3: Code for reproducibility.
>
> A3: We followed the guideline of ICLR to provide the link of our code in the Reproducibility Statement, which is right after the conclusion section in our paper.
>
> C4: How do authors compute truncated SVD within 7 mins on OGB datasets?
>
> A4: We used “svds” from scipy. Our hardware information is available in Appendix A.9.3.
>
> C5: Mini-batch training or full batch training on OGB graphs?
>
> A5: We perform full batch training on all experiments. More information about experimental setup is available in Appendix A.9.

---

> > ### Comment · Reviewer_EL8y · 2021-12-01
> > **Many Thanks to Your Feedback**
> >
> > Thank you for the detailed and thoughtful feedback. I have also read the comments from other reviewers as well as the corresponding feedbacks.
> >
> > Most of my concerns are addressed, especially the supplementary computational complexity analysis and further details regarding to reproducibility are interesting and helpful.
> >
> > On the other hand, I am not convinced on the response towards an adaptive attack, which would be more of interest and needs further exploration and discussion in my opinions. Thus, I would like to keep my score.

---

### Official Review · Reviewer_HF71 · 2021-11-03

**Correctness:** 2
**Technical Novelty And Significance:** 2
**Empirical Novelty And Significance:** 2
**Recommendation:** 3
**Confidence:** 4

**Details Of Ethics Concerns:**

There is no ethics concern.

**Main Review:**

## Strengths

1. The paper introduces an efficient reduced-rank approximation method to purify the graph structure that is scalable to large graphs.

2. It provides theoretical justification for the matrix sparsification method.

3. The paper presents lots of experiments to demonstrate the robustness of the proposed method in different settings, such as targeted and untargeted attacks, homophilic and heterophilic graphs, small and large graphs, etc.


## Weaknesses

1. The proposed GARNET combines several techniques proposed in previous works such as Low-rank approximation in GCN-SVD and ProGNN, adaptive filter learning in GPR-GNN, as well as correct and smooth in C&S method. Overall, the novelty of the proposed method is quite limited, and the contribution of this paper is unclear.

2. The comparison with the baseline models doesn’t seem to be fair. From my understanding, although the baseline methods (such as Pro-GNN), use classic 2-layer GCN as the backbone model, according to their formulation, their ideas can be generally applied to more advanced GNN models such as APPNP and GPRGNN as well. In essence, GARNET in this paper chooses a more advanced and deeper backbone model that adopts adaptive filter as in GPRGNN and adaptive label propagation as in C&S to compare with the baselines which only use the basic GCN as their backbone models. This doesn’t seem to be a fair comparison since the depth (e.g., number of propagation layers) and filter design typically have significant impacts on the robust performance. Therefore, the performance improvement is not convincing.

3. As far as I am concerned, the low-rank approximation idea is essentially similar to existing works and this paper proposes a more efficient way to achieve this. Therefore, regardless of the efficiency and scalability, the robust performance should be similar to existing works if appropriate backbone models are chosen in the comparison. It is better to have a discussion on this.

4. From the ablation study in Section 4.4, Figure 2 only compares GARNET and GNNGuard. However, GNNGuard is less relevant in this context and the performance is much worse than Pro-GNN in this setting as showed in Table 2 or Table 3. It would be more interesting to see how the low-rank approximation in this work improves over Pro-GNN instead of GNNGuard.




**Summary Of The Paper:**

The paper proposes a reduced-rank and sparse approximation method to purify the graph structure for better robustness against adversarial graph attacks. The approximation is accelerated through spectral embedding and sparsification. Moreover, adaptive graph filter and label propagation are considered for further improvement on low homophily graphs.



**Summary Of The Review:**

Overall, the contributions of the paper are unclear and the novelty seems to be limited. In addition, the comparison and ablation study in the experiment is not totally convincing. I would like to suggest a rejection.

---

> ### Author Response · Authors · 2021-11-16
> **Additional Experiments and Clarifications**
>
> Thanks for your detailed review. We submitted the revised paper according to your comments. We are going to address your concerns one by one as follows:
>
> C1: Low-rank, adaptive filter, and C&S are not novel.
>
> A1: (1) Contribution of reduced-rank kernel: our approach is better than the existing SVD-based methods in terms of both scalability and adversarial accuracy. For scalability, we avoid computing/storing a dense adjacency matrix that is required in those SVD-based methods. Thus, our method can scale to a large graph with millions of nodes, while prior SVD-based methods (e.g., ProGNN) run out of memory on our hardware even on a graph with 10k nodes (e.g., Pubmed). For the accuracy contribution, please refer to our response “A3” below.
> (2) Contribution of adaptive filtering and label propagation kernels: although the notion of adaptive filtering is not novel, we are the first one to theoretically show adaptive filtering works under the adversarial setting when combined with our reduced-rank graph in Theorem 2. In addition, to our knowledge, we are also the first one to leverage the learned adaptive filter to guide the label propagation process, which is a simple yet effective way to extend existing label propagation methods (e.g., C&S) to work for graphs with different levels of homophily.
>
> C2: Comparison with baselines is not fair. Authors should compare with more advanced backbone models such as SVD+GPRGNN+CS.
>
> A2: We admit that we should compare our method to baseline methods with more advanced models. Note that the vanilla ProGNN with only 2-layer GCN already takes more than 24 hours for 10 runs on Cora with 3k nodes, and runs out of memory on Pubmed. Thus, integrating more advanced models into ProGNN would make the training time too slow or even out-of-memory since ProGNN trains on a dense adjacency matrix from scratch. Instead, we make additional comparisons with SVD+GPRGNN+CS (i.e., GPRSVD-CS). The new results are added to our revised paper (marked in blue in Tables 1-3). GARNET still improves accuracy over GPRSVD-CS by a margin up to 9.7% and 18.06% on homophilic and heterophilic graphs, respectively. We attribute such a large accuracy improvement to the advantage of our reduced-rank kernel over TSVD. Please refer to our response “A3” below for details.
>
> C3: Why is GARNET much better than GCNSVD?
>
> A3: The reason is that TSVD aggressively reduces the graph rank which inevitably loses important spectral information. Specifically, TSVD typically chooses top k largest singular values and their corresponding singular vectors with a relatively small k (e.g., k=50) for efficient computation, which means the resulting low-rank adjacency matrix only has the rank of 50. As shown in Figure 5 (Appendix A.8), however, the input graph has the rank>2200 on Cora. Thus, the 50-rank graph obtained by TSVD loses too much information in the graph spectrum.
>
> In contrast, Figure 5 shows our reduced-rank method only removes the highest singular components from the graph and produces a kNN graph with rank~=1900 on Cora, which preserves the key spectral information that contributes to the GNN training. In other words, our reduced-rank approach not only tackles the scalability issue, but also improves the accuracy over the TSVD-based methods by preserving much more useful information from the graph spectrum.
>
> We highlight the advantage of our reduced-rank method over SVD in our revised paper (text marked in blue in Section 3.1) and show the difference between these two algorithms in Appendix A.11. Moreover, we empirically confirm this by comparing GCN-SVD with GCN-RRA (GCN+our reduced-rank approximation kernel) in Appendix A.11. The results show that GCN-RRA largely outperforms GCN-SVD with accuracy improvement up to 15.08% (see Table 9), revealing that our reduced-rank graph leads to a much higher accuracy than that of SVD-based defense models.
>
> C4: Ablation study should compare with Pro-GNN instead of GNNGuard.
>
> A4: We have added the results of ProGNN in ablation study to our revised paper (see Figure 2). Note that Pro-GNN performs poorly on low heterophilic graphs (e.g., Squirrel), since it assumes feature smoothness for training the model.

---

> > ### Comment · Reviewer_HF71 · 2021-11-30
> > **New comment**
> >
> > Thank you for the response!

---

> > > ### Author Response · Authors · 2021-11-30
> > > **Addressed your concerns?**
> > >
> > > Dear Reviewer HF71,
> > >
> > > Thank you for your reply. We are wondering if our responses have successfully addressed your major concerns?

---

> > > > ### Comment · Reviewer_HF71 · 2021-11-30
> > > > **New comments**
> > > >
> > > > While I still believe that the adaptive filter and the C&S method are not the contributions of this paper, I consider the reduced-rank approximation as the major contribution of this paper.
> > > >
> > > > However, from your answer, it is still not clear to me why the proposed reduced-rank approximation can significantly outperform truncated SVD. There is a lack of motivation or intuition for such improvements.
> > > >
> > > > The motivation that drives you to design the sparsified approximation is to improve the efficiency of using the dense low-rank adjacent matrix. The approximation is based on the top-R truncated SVD (e.g., R=50), which already removes most of the high singular components of the graph. Essentially, the proposed approximation just specifies the top-R truncated SVD in an efficient way. The sparsification might increase the rank of the matrix again (as you showed), but this does not maintain the original spectral information considering that it is from the truncated SVD.
> > > >
> > > > It is claimed that "the reduced-rank method only removes the highest singular components, while retaining most of important spectral information". However, there is a lack of intuition or understanding about what kind of additional graph information the specification process brings beyond the truncated SVD.  What are those "important spectral information" and why they are useful?

---

> > > > > ### Author Response · Authors · 2021-11-30
> > > > > **Additional Experiments and Clarifications**
> > > > >
> > > > > Thank you for your further questions. We hope our new responses shown below can address your remaining concerns.
> > > > >
> > > > > C1: Intuition of sparsification for improving adversarial accuracy.
> > > > >
> > > > > A1: We have new results showing that the adversarial accuracy of GCN-SVD increases from $70.36$ to $78.96$ if we increase the graph rank from $50$ to $1900$ by using more singular vectors/values via running truncated SVD on Cora, which means increasing the rank of graph can indeed largely improve adversarial accuracy. However, computing such a large number of singular vectors/values is computationally expensive. Since using more singular vectors/values typically makes the constructed graph sparser (the extreme case is that we use all the singular vectors/values and thus we recover the input sparse graph), we use our sparsification approach to increase the graph rank and thus improve adversarial accuracy. We will add our latest results and explanation in the next revision.
> > > > >
> > > > > C2: Maintaining original spectral information via using our weighted spectral embedding.
> > > > >
> > > > > A2: Our weighted spectral embedding step can be viewed as an efficient way of structure preserving embedding [1], which can be used to recover original edges (or spectral components) via kNN method. We also empirically confirm this by showing GCN-SVD (with rank of 1900 by using top 1900 singular vectors/values) and GCN-RRA (our reduced-rank method with graph rank ~= 1900) achieve similar adversarial accuracy ($78.96%$ and $79.75%$ respectively) on Cora dataset, which means our RRA method can preserve similar spectral information of those top 1900 singular vectors/values from the original (input) graph for GNN training.
> > > > >
> > > > > C3: What are those spectral information and why they are useful?
> > > > >
> > > > > A3: As shown in [2], adversarial attacks only perturb the top few highest singular components in the graph spectrum, while most of the spectral information corresponds to the clean graph structure in the spatial domain. Thus, our approach can preserve the clean graph structure that contributes to the GNN training.
> > > > >
> > > > > Reference
> > > > >
> > > > > [1] Shaw, Blake, et al. "Structure Preserving Embedding." ICML 2009
> > > > >
> > > > > [2] Entezari, Negin, et al. "All you need is low (rank) defending against adversarial attacks on graphs." WSDM 2020.

---

### Official Review · Reviewer_F9Qe · 2021-11-04

**Correctness:** 3
**Technical Novelty And Significance:** 2
**Empirical Novelty And Significance:** 2
**Recommendation:** 5
**Confidence:** 4

**Main Review:**

Strengths

+ The paper is clearly written. All the technical steps of GARNET are easy to understand.
+ The paper presents extensive empirical evaluation on the proposed framework. The datasets cover both homophily and heterophily graphs, and from small to large scales.


Weaknesses

- Although the three main steps of GARNET are all reasonable, they lack some technical significance. For example,
    - The low rank approximation step follows a standard algorithm and the description over-complicates the algorithm. Although the authors call the reconstructed graph a "k-NN" graph, it is indeed just obtained by row-wise preserving the largest elements in the rank-r approximate adjacency matrix -- according to Thm 1, the similarity score used by k-NN is exactly the element value of the rank-r adj matrix. The spectral embedding matrix is also well-known (e.g., used in spectral clustering), and the authors are not really utilizing much property of the spectral embedding except deriving the low-rank approximation.
    - The adaptive filtering step is also not novel. Multiple existing works follow similar form of such filtering (e.g., SIGN [a]).
    - Theorem 2 does not seem useful, since the upper bound is way too loose. If we compare the upper bound with the scale of the original perturbation, then the derived upper bound is even orders of magnitude larger. If the upper bound is achieved, then it would tell us that adaptive filtering can enlarge the effect of perturbation. In addition, the upper bound grows quadratically with filtering depth P, indicating that heterophily graphs could be problematic (from theoretic perspective).
- While the experiments show significant accuracy gains, further clarification could make the improvements more convincing.
    - In Table 2, 3 and 4, we can see that GARNET achieves higher accuracy even on the clean graph. I suppose this is due to the adaptive filtering step. However, since adaptive filtering is a known design, it would be better to add one unvaccinated baseline model following Eq 2 and 3 in these tables.
    - From Fig 2, it is clear that reducing the rank is the primary reason for successful defense. If this is the case, then it is not clear why similar rank-reduction based method, GCNSVD, achieves much lower performance than GARNET in Tables 2 and 3. More explicitly, since in Fig 2, GNNGuard achieves much lower accuracy than GARNET, I would imagine GCNSVD to achieve much higher accuracy than GNNGuard and slightly lower accuracy than GARNET in Tables 2 and 3. Yet this does not seem to be the case.


[a]: SIGN: Scalable Inception Graph Neural Networks, 2020.

**Summary Of The Paper:**

This paper proposes GARNET, a spectral approach to depend adversarial attacks on graph structure. GARNET consists of three steps, low rank approximation of adjacency matrix, adaptive filtering on node signals and label propagation. All the 3 steps of GARNET achieve low computation complexity and thus can be applied to large graphs. Extensive experiments on both homophily and heterophily graphs show that GARNET achieve significant accuracy improvements compared with baselines, when the structural perturbation on graphs are stronger.

**Summary Of The Review:**

I think the paper is lower than the acceptance threshold due to the lack of technical significance and question in empirical evaluation, as detailed above.

---

> ### Author Response · Authors · 2021-11-16
> **Additional Experiments and Clarifications**
>
> Thanks for your detailed review. We submitted the revised paper according to your comments. We are going to address your concerns one by one as follows:
>
> C1: The description of low-rank approximation over-complicates the algorithm.
>
> A1: Our kNN-based method is different from naively preserving large elements in r-rank matrix A in terms of scalability. Note that preserving the largest elements of each row in the r-rank matrix A requires forming/storing A first, which has quadratic time/space complexity and is not scalable to large graphs, since A is a dense matrix obtained by TSVD. In contrast, our reduced-rank graph is based on an approximate kNN, and it does not require forming/storing any dense matrices. Hence our approach can scale to large graphs with millions of nodes, while previous state-of-the-art baselines (e.g., ProGNN, GCNSVD) cannot. Currently, the way that we describe is exactly how we can improve the scalability while reducing the graph rank. We will try to simplify the description and better articulate our key advantages in the next revision.
>
> C2: Not utilizing much property of spectral embeddings.
>
> A2: Conventional spectral embeddings (extensively studied in spectral graph theory) only leverage the first few (smallest) nonzero eigenvalues and their corresponding eigenvectors of the graph Laplacian matrix. In contrast, we use the top few largest singular values and their corresponding singular vectors of the adjacency matrix, which not only effectively reduces the rank of the adjacency matrix, but also improves the approximation quality of the reduced-rank graph. Notably, these singular components correspond to both the smallest and largest eigenpairs of the Laplacian matrix, which allow retaining both the global and local structural properties, respectively. We have added new results about the comparison between the classical spectral embedding method and the proposed method in Appendix A.10, showing our singular components based spectral embedding leads to much better accuracy than the conventional spectral embedding.
>
> C3: The adaptive filtering is not novel.
>
> A3: While there are prior arts using adaptive filtering techniques, we are the first to show adaptive filtering also works under adversarial settings when combined with our reduced-rank graph (see Theorem 2). In addition, to our knowledge, we are also the first to leverage learned adaptive filters to guide the label propagation process, which is a simple yet effective way to extend existing label propagation methods (e.g., C&S) to work for graphs with different levels of homophily.
>
> C4: The upper bound of Theorem 2 is loose.
>
> A4: Theorem 2 aims at showing the upper bound of the difference between $P$-degree polynomial filters, which is directly related to the difference of filtered node embeddings, rather than the difference between the adjacency matrices. Note that the difference between $P$-degree polynomial filters (i.e., left hand side of Equation 4) also becomes larger than the scale of original perturbation when increasing the $P$ value. Nevertheless, we agree that the term $P^2$ may make our upper bound loose for a large $P$ (e.g., $P=10$).  Thus, we provide a tighter upper bound in our revised paper (marked in blue) and the proof is available in Appendix A.2. Note that we enforce $P<=10$ and $\sum_{p=0}^P c_p = 1$ for training the adaptive filter. Suppose $c_p = \frac{1}{P+1} \forall p=0,1,...,P$, the upper bound then becomes $\frac{P}{2} (\epsilon + \sigma_{r+1})$, which no longer grows quadratically with $P$.
>
> C5: Adding the comparison to unvaccinated adaptive filtering learning (GPRGNN) model.
>
> A5: We have added new results of GPRGNN to Tables 1-3 in our revised paper (colored in blue). The results show that GARNET consistently outperforms GPRGNN by a margin up to 12.53%.
>
> C6: It is unclear why GARNET is much better than TSVD-based models (e.g., GCNSVD).

---

> > ### Author Response · Authors · 2021-11-16
> > **Continued anwser**
> >
> > A6: The reason is that TSVD aggressively reduces the graph rank which may inevitably lose important graph structural information. Specifically, TSVD typically chooses top k largest singular values and their corresponding singular vectors with a relatively small k (e.g., k=50) for efficient computation. Hence the resulting low-rank adjacency matrix only has the rank of 50. As shown in Figure 5 (Appendix A.8), however, the input graph of Cora has a rank>2200. Thus, the 50-rank graph obtained by TSVD loses too much information in the graph spectrum; this is why it even has a lower accuracy than non-rank-reduced models (e.g., GNNGuard).
> >
> > In contrast, Figure 5 shows our RRA method only removes the highest singular components from the graph and produces a kNN graph with rank~=1900 on Cora, which preserves the key spectral information that contributes to the GNN training. In other words, our reduced-rank approach not only tackles the scalability issue, but also improves the accuracy over the TSVD-based methods by preserving much more useful information from the graph spectrum.
> >
> > We highlight the advantage of our reduced-rank method over SVD in our revised paper (text marked in blue in Section 3.1) and show the difference between these two algorithms in Appendix A.11. Moreover, we empirically confirm this by comparing GCN-SVD with GCN-RRA (GCN+our reduced-rank approximation kernel) in Appendix A.11. The results show that GCN-RRA largely outperforms GCN-SVD with accuracy improvement up to 15.08% (see Table 9), revealing that our reduced-rank graph leads to a much higher accuracy than that of SVD-based defense models.

---

> > > ### Comment · Reviewer_F9Qe · 2021-12-01
> > > **Post Rebuttal Comments**
> > >
> > > I thank the authors for their detailed responses. I have read them carefully. In summary, the response has partially addressed my concerns, but some of my major concerns still remain. I would like to keep my original score.
> > >
> > > Specifically,
> > >
> > > C1: thanks for the clarification. Now I understand that k-NN step is essential to avoid the N^2 complexity. Then the description in the technical section seems appropriate. Yet, from model perspective, k-NN + thresholding still aims at keeping the largest value. Besides, approximate k-NN algorithm is known. So it is hard to claim much novelty here.
> > >
> > > C2: some technical details may be different between classic spectral embedding and GARNET. However, I still feel that the key idea is very similar. Essentially, GARNET is still doing the low rank approximation. But because of the approximate algorithm (e.g., k-NN), the model can afford to preserve more information on large graphs. In addition, it is not surprising that GARNET keeps both the smallest and largest eigenvalues -- singular values on symmetric matrix equals the absolute value of the eigenvalues. So smallest, negative eigenvalues are / should be naturally preserved.
> > >
> > > C3: while it is good to show that you are the first to apply adaptive filtering to adversarial setting, in my opinion "being the first to do something" does not imply "novelty". It still seems to me a quite straightforward combination.
> > >
> > > C4: thanks for tightening the bound. Yet, most of my original comments are still valid. P-propagation still enlarges the perturbation. So in my opinion, the bound is still too loose to be useful.
> > >
> > > C5, C6: thanks for these additional experiments. The results look promising.

---

### Public Comment · ~Benedek_Andras_Rozemberczki1 · 2021-11-14
**Chameleon and Squirrel Dataset Attribution**

The datasets were introduced in this paper:

@article{rozemberczki2021multi,
  title={Multi-scale attributed node embedding},
  author={Rozemberczki, Benedek and Allen, Carl and Sarkar, Rik},
  journal={Journal of Complex Networks},
  volume={9},
  number={2},
  pages={cnab014},
  year={2021},
  publisher={Oxford University Press}
}

---

> ### Author Response · Authors · 2021-11-16
> **Dataset Citation**
>
> Thanks for pointing that out. We have cited the paper in our revision.

---

### Author Response · Authors · 2021-11-16
**Summary of Revision**

Thanks for the constructive comments from all reviewers. We have included additional experimental results and clarification into our revision (changes marked in blue). We highlighted the contribution of our reduced-rank approximation (RRA) method, and explained why GARNET achieves higher accuracy than defense methods based on truncated SVD (see Section 3.1). In our evaluation, we added comparisons with GPRGNN and GPRSVD-CS (GPRGNN+SVD+C&S) as stronger unvaccinated and vaccinated baselines, respectively (see Tables 1-3). In addition, we added Pro-GNN and GPRSVD-CS as baselines in our ablation study in Figure 2. These experiments provide further evidence that GARNET significantly improves adversarial accuracy over prior arts. We hope the new clarification of our contribution and empirical results address the main concerns of Reviewer F9Qe and HF71. We also provided the overall space/time complexity analysis of GARNET in Appendix A.3. We hope this clears the doubts raised by Reviewer EL8y.

---

### Decision · Program_Chairs · 2022-01-20

**Decision:**

Reject

**Comment:**

The paper proposes a method to change the graph structure for better robustness against adversarial attacks. The reviewers commend the authors for a clearly written paper and promising results. Several reviewers expressed concerns about experimental validation (specifically, comparison to truncated SVD and choice of baselines), complexity, and novelty. The rebuttal and follow-up discussion alleviated some of the concerns, but the reviewers still have outstanding issues, therefore the AC does not recommend accepting the paper.